# ONLINE PREDICTION OF STOCHASTIC SEQUENCES WITH HIGH PROBABILITY REGRET BOUNDS

**Matthias Frey, Jonathan H. Manton, and Jingge Zhu**
Department of Electrical and Electronic Engineering, The University of Melbourne

## ABSTRACT

We revisit the classical problem of universal prediction of stochastic sequences with a finite time horizon $T$ known to the learner. The question we investigate is whether it is possible to derive vanishing regret bounds that hold with high probability, complementing existing bounds from the literature that hold in expectation. We propose such high-probability bounds which have a very similar form as the prior expectation bounds. For the case of universal prediction of a stochastic process over a countable alphabet, our bound states a convergence rate of $\mathcal{O}(T^{-1/2}\delta^{-1/2})$ with probability as least $1 - \delta$ compared to prior known in-expectation bounds of the order $\mathcal{O}(T^{-1/2})$. We also propose an impossibility result which proves that it is not possible to improve the exponent of $\delta$ in a bound of the same form without making additional assumptions.

## 1    INTRODUCTION

A classical question in online learning, information theory, and several related fields is how to make predictions about the outcome $Z_t$ of a process at time $t$ given some knowledge of the past outcomes $Z_1, \ldots, Z_{t-1}$. Its appearance in the early works by Shannon (1951); Bellman (1954); Hannan (1957) reflects the central role it plays in several pure and applied research areas of mathematics, computer science, and engineering. This basic question comes in many variations such as: (i) The process $(Z_t)$ can follow a fixed, but unknown probability distribution, or it can be deterministic and unknown, which even includes the possibility that it is chosen adversarially given the learner's strategy (Merhav & Feder, 1998). (ii) The learner can have access to the advice of experts (Weissman & Merhav, 2001) or other side information about the process, or it can make its predictions without access to such information (Bellman, 1954). (iii) The learner can make its predictions only for $Z_1, \ldots, Z_T$, where $T$ can be a fixed time horizon known to the learner (Bellman, 1954) or a random stopping time (Morvai & Weiss, 2007); or the learner can make predictions indefinitely for $(Z_t)_{t \in \mathbb{N}}$ (Gyorfi et al., 1999). (iv) The learner can have access only to past outcomes $Z_1, \ldots, Z_{t-1}$ (Bellman, 1954), or it can have access to an infinite past $(Z_{t'})_{t'=-\infty}^{t-1}$ (Gyorfi et al., 1999).

In this paper, we focus on a particular one of these scenarios, which is, however, still general enough to be considered a fundamental question. Namely, we assume that the learner makes predictions about a random process $Z_1, \ldots, Z_T$ where $T$ is a deterministic time horizon known to the learner. We further assume that $Z_1, \ldots, Z_T$ follows a joint probability distribution $P$ which does not need to have identical or independent components, but is not influenced by the predictions $(B_t)_{t=1}^T$ the learner makes at each time instant $t$ based on full knowledge of $Z_1, \ldots, Z_{t-1}$ (we use the common convention in this paper that $Z_1, \ldots, Z_0$ is the empty sequence). The quality of the learner's decisions is measured in terms of a loss $\ell(B_t, Z_t)$. The main quantity of interest is the *regret* of the learner in hindsight, i.,e., the difference between the cumulative loss incurred by the learner and the cumulative loss incurred by a prediction strategy that would have been optimal given full knowledge of $P$. This scenario has been proposed in this form by Merhav & Feder (1998) and solved *in expectation*: that is, the authors analyze the expectation of the learner's regret. To the best of our knowledge, our paper is the first that studies *high-probability bounds* of the learner's regret for this particular scenario, i.e., we propose bounds that the learner's regret surpasses only with a small probability $\delta$. Such high-probability bounds hold enormous promise for practical applications where reliability is key, such as accident prediction in air traffic control (Amin et al., 2024), trajectory prediction in autonomous driving (Zhang et al., 2022), and sepsis prediction in health care (Boussina et al., 2024).

The contribution of this paper can be summarized as following:

- For the problem of universal prediction of stochastic sequences under a general, bounded loss, we propose a high-probability regret bound (Theorem 4) that complements existing in-expectation bounds in the literature. Our high-probability bound matches the prior expectation bound in terms of convergence order in the time horizon $T$. To the best of our knowledge, high-probability bounds for this scenario have not appeared in previous works.

- We show with an impossibility result (Theorem 5) that our proposed high-probability regret bound cannot be significantly improved regarding the order of the permissible error probability that appears in the bound without making additional assumptions.

- In contrast with Merhav & Feder (1998), our results do not necessarily require that the underlying spaces are finite. Instead, we use much milder technical assumptions.

The remainder of the paper is structured as follows: In Section 2, we survey relevant prior works on the topic of prediction of sequences. In Section 3, we introduce the universal prediction problem and give a high-level overview of our high-probability result and compare it to the most closely related expectation bound found in prior works. We fully formalize a simplified version of the problem, which we call the mismatched prediction problem, in Section 4 and introduce all notations and definitions that are necessary to state and discuss our results in Section 5. In Section 6, we show how our result on mismatched prediction can be applied in conjunction with known techniques from the literature to yield high-probability convergence rates for the universal prediction problem. In Section 7, we describe the numerical experiments we have run to illustrate the promise of the schemes studied in this paper for practical applications, and we report on their results. The paper is concluded in Section 8 with a discussion of limitations and future research directions.

## 2 RELATED WORKS

The wider question of prediction of sequences is a classical research area. The distinct but related bandit problem has first been studied by Thompson (1933) (see, e.g., Lattimore & Szepesvári (2020) for an overview of more recent developments). A scenario that is very closely related to the one we study in this paper was to the best of our knowledge first described by Shannon (1951). He posed the question of how well a stochastic process could be predicted given past observations and related the prediction quality to an information quantity, namely what is now called Shannon entropy. He used this connection to estimate the entropy contained in expressions in the English language based on how accurately a human test subject would be able to predict the next letter in an incomplete sentence. The first works concerned with strategies for optimum prediction of future realizations of a sequence were Bellman (1954); Hannan (1957) for the case of a deterministic process unknown to the learner (and potentially selected by an adversary with knowledge of the learner's strategy). The further development of the various flavors of prediction of processes throughout the remainder of the 20th century is summarized by Merhav & Feder (1998) which is also the earliest work we are aware of that proposes the exact version of the problem which we study in this work (but provides bounds only for *expected* regret).

**Prediction with expert advice.** In this version of the problem, the learner has access to advice given by so-called *experts*, where at each time step, it can decide which expert's advice to follow. The regret in this case is typically the difference between the loss incurred by the learner and the loss incurred by the best available expert in hindsight. Examples of works considering this version of the problem include Cesa-Bianchi & Lugosi (2001); Weissman & Merhav (2001); Wu et al. (2023b); Hanneke et al. (2023) for deterministic sequences and Weissman & Merhav (2004) for stochastic sequences.

**Prediction of deterministic sequences.** The most prevalent version of the sequential prediction problem does not make any stochastic assumptions about the underlying process that has to be predicted and studies bounds on the worst-case regret. That is, the predicted sequence is assumed deterministic and could in principle be chosen by an adversary with knowledge of the learner's prediction strategy. Early works on this include Bellman (1954); Hannan (1957), the problem is also studied and surveyed by Merhav & Feder (1998), and a very comprehensive treatment can be found in Cesa-Bianchi & Lugosi (2006). More recent developments include Abernethy et al. (2012); Hanneke et al. (2023); Cutkosky & Mhammedi (2024).

It is worth noting that an unknown deterministic sequence can be viewed as a stochastic sequence that follows an unknown Dirac distribution. So results for the prediction of stochastic sequences can potentially be relevant for deterministic sequences as well. However, if the underlying sequence follows a Dirac distribution, this means in particular that if the learner's predictions are deterministic conditioned on the part of the sequence that has already been observed (as is the case in the strategy analyzed in this paper), the instantaneous regret is almost surely equal to the average regret. Hence, our high-probability results do not offer any advantage over the known bounds in expectation in the case of deterministic sequences.

**Prediction of stochastic sequences.** Under the assumptions of access to an infinite past and stationarity, ergodicity, or Markovness of the underlying process, this problem was studied by Algoet (1994); Gyorfi et al. (1999); Gyorfi & Ottucsak (2007) (see Morvai & Weiss (2007) for a survey of more results along these lines). The assumption of access to an infinite past was lifted by Morvai & Weiss (2011) who proved almost sure convergence of regret quantities, but did not provide quantitative high-probability bounds. Bounds on expected regret were studied by Merhav & Feder (1998), Hutter (2003), Watanabe & Roos (2015) (under an i.i.d. assumption), and Wu et al. (2023a) (with the additional complication that the underlying process may change its distribution a certain number of times). High-probability bounds for stochastic scenarios have been investigated by Mehta (2017) (under the assumption of a log-concave loss and an i.i.d. underlying process) and van der Hoeven et al. (2023) (for an online-to-batch conversion scenario, i.e., it is shown that high-probability bounds on regret can imply high-probability bounds on generalization error of a batch version of the learner). The very recent works Ghosh et al. (2024); Qin et al. (2025) deal with a signal processing application of the sequential prediction problem and employ recurrent neural networks (RNNs) in their predictors.

To the best of our knowledge, there are no prior works that study high-probability bounds for the regret in prediction of non-i.i.d. stochastic sequences in the absence of expert advice.

## 3 THE UNIVERSAL AND THE MISMATCHED PREDICTION PROBLEMS

In the universal prediction problem with stochastic sequences (Merhav & Feder, 1998, Section I-A), the learner observes the initial part $Z_1, \ldots, Z_{t-1}$ of some sequence $(Z_1, \ldots, Z_T) \in \mathcal{Z}^T$ which follows a joint probability distribution $P$. After it makes a prediction $B_t \in \mathcal{B}$, the next realization $Z_t$ is revealed, and the learner incurs a loss $\ell(B_t, Z_t)$. The objective is to find a prediction strategy which is *universal* in the sense that it does not depend on $P$ (and hence requires no knowledge of it) but still approaches — for large enough $T$ — the performance of a strategy that is optimized for $P$. The performance is evaluated in the following way: we consider the difference between the average loss the learner's strategy incurs up to time $T$ and the average loss that a strategy optimized for $P$ incurs up to time $T$. This difference is called the *regret* and denoted $\Delta$ (for a formal definition see (4) and (5) below). A universal strategy will have vanishing $\Delta$ as $T$ grows, regardless of what distribution $P$ (within a certain known class of probability distributions) the sequence $Z_1, \ldots, Z_T$ follows.

It is often assumed that $P = P_\theta$ for some $\theta \in \Theta$ where $\Theta$ is a measurable space and $(P_\theta)_{\theta \in \Theta}$ is a parametrized family of probability distributions known to the learner. The idea in case very little is known about $P$ is to use a family $(P_\theta)_{\theta \in \Theta}$ that is so large that it will contain (almost) all $P$ that are reasonably possible. One example of theoretical interest is investigated by Hutter (2003): let $\Theta$ be the set of all programs on a universal Turing machine that compute a probability distribution on $\mathcal{Z}^{\mathbb{N}}$ to arbitrary precision, and let $P_\theta$ be the distribution computed by $\theta \in \Theta$. Then, any prediction strategy that is universal for $(P_\theta)_{\theta \in \Theta}$ guarantees vanishing regret for every sequence $Z_1, \ldots$ that follows a probability distribution which is computable (in the sense that there exists a program on a universal Turing machine that computes it to arbitrary precision). This is not directly applicable to practical problems, of course, but it is a very powerful theoretical illustration of a basic idea that can also be useful in practice: based on very limited knowledge of properties of $P$, make the family $(P_\theta)_{\theta \in \Theta}$ so large that it can be guaranteed it contains $P$. We do not make a contribution about how $(P_\theta)_{\theta \in \Theta}$ can be chosen and how such a family induces a universal prediction strategy, but since this is essential to understand the consequences of our results, we briefly survey the literature on this in Section 6.

We next give an idea of the nature of the implications of our results by informally stating a prior result on universal prediction and one of the consequences of our technical results.

**Theorem 1** (Merhav & Feder (1998); Hutter (2003)). *If $\mathcal{Z}$ is countable, there is a strategy which is universal for all computable $P$ in the sense that we have $\mathbb{E}\Delta \leq \mathcal{O}(1/\sqrt{T})$.*

While this is the best rate result for general bounded losses to the best of our knowledge, it is for instance possible in the case of self-information loss to get the better rate $\mathcal{O}(1/T)$ (Merhav & Feder, 1998; Hutter, 2003).

Our main results imply the following high-probability version of Theorem 1:

**Theorem 2.** *If $\mathcal{Z}$ is countable, there is a strategy which is universal for all computable $P$ in the sense that for every $\delta \in (0,1)$, we have $P\left(\Delta < \mathcal{O}(1/\sqrt{T\delta})\right) \geq 1 - \delta$.*

Theorem 2 follows from (18) of Theorem 4 proposed in Section 5 in conjunction with the considerations by Hutter (2003) sketched above and discussed in more detail in Section 6. The simplifying assumption that $\mathcal{Z}$ is countable can be replaced by alternative assumptions. Therefore, we have similar results for some cases in which $\mathcal{Z}$ is a general measurable space. However, this comes at a small cost in terms of order of convergence. For details, see Section 6 and Table 1 contained therein.

In the technical problem statement in Section 4 and our main results in Section 5, we consider a simplified problem which we call *mismatched prediction*. In this scenario, we assume that the learner's strategy can be mathematically represented as a probability measure $Q$ (which is mismatched to the problem in the sense that it is possibly different from $P$, but is still a "reasonable" approximation of it) and analyze the regret incurred by such strategies. This is the same stepping stone used by Merhav & Feder (1998) to tackle the universal prediction problem, and results for the universal prediction problem follow from results for mismatched prediction via known results from the literature.

Specifically, the distribution $Q$ is usually chosen as $Q := \int_\Theta P_\theta w(d\theta)$, where $(P_\theta)_{\theta \in \Theta}$ is a parametrized family of all distributions that are "reasonably possible" and $w$ is a probability distribution on $\Theta$. $w$ does not need to accurately represent some prior knowledge of the distribution of $w$, in fact it is sufficient for Theorems 1 and 2 to hold that $P = P_\theta$ for some $\theta$ in the support of $\Theta$. Therefore, it is common in practice to choose $w$ as a uniform distribution when possible. In Section 6, we return to the universal prediction problem, formalize it fully, and explain how results for universal prediction follow from results for mismatched prediction.

## 4 NOTATIONS AND TECHNICAL STATEMENT OF THE MISMATCHED PREDICTION PROBLEM

Let $(\Omega, \mathscr{F}, \mathbb{P})$ be a probability space and $Z_1, \ldots, Z_T$ be (not necessarily independent) random variables each taking values in a measurable space $(\mathcal{Z}, \mathscr{G})$. The joint distribution of $Z_1, \ldots, Z_T$ is described by a probability measure $P$ on $(\mathcal{Z}^T, \mathscr{G}^{\otimes T})$ where $\mathscr{G}^{\otimes T} := \mathscr{G} \otimes \cdots \otimes \mathscr{G}$ denotes the product $\sigma$-algebra with $T$ copies of $\mathscr{G}$ as factors. For $\mathcal{T} \subseteq [T] := \{1, \ldots, T\}$, we denote with $P_\mathcal{T}$ the marginal distribution of $(Z_t)_{t \in \mathcal{T}}$. We assume that for every $t \in [T] \setminus \{1\}$, the distribution $P$ admits a Markov kernel $K_P^{(t)} : \mathcal{Z}^{t-1} \times \mathscr{G} \to [0, 1]$ such that for all $G_1, \ldots, G_t \in \mathscr{G}$

$$P_{[t]}(G_1 \times \cdots \times G_t) = \int_{G_1 \times \cdots \times G_{t-1}} K_P^{(t)}(z_1, \ldots, z_{t-1}, G_t) P_{[t-1]}(dz_1, \ldots, dz_{t-1}). \quad (1)$$

We further adopt the convention $K_P^{(1)} := P_1$ and the convention that integration over a tuple of length 0 returns the integrand, so that (1) also holds for $t = 1$.

**Problem 1** (Prediction of stochastic sequences). *The prediction problem $\mathcal{P} = (T, (\mathcal{Z}, \mathscr{G}), P, (\mathcal{B}, \mathscr{H}), \ell)$ is characterized by the time horizon $T \in \mathbb{N}$, a measurable space $(\mathcal{Z}, \mathscr{G})$, a probability measure $P$ on $(\mathcal{Z}^T, \mathscr{G}^{\otimes T})$ which describes the joint distribution of a tuple $Z_1, \ldots, Z_T$ of random variables, a measurable space $(\mathcal{B}, \mathscr{H})$ (the prediction domain), and a measurable[1] loss function $\ell : \mathcal{B} \times \mathcal{Z} \to [0, L]$, where $L \in [0, \infty)$. The learner's policy $b = (b_t)_{t \in [T]}$ consists of measurable (deterministic) functions $b_t : \mathcal{Z}^{t-1} \to \mathcal{B}$.*

*Nature first draws a tuple $Z_1, \ldots, Z_T$ randomly from the distribution $P$. At each time $t \in [T]$:*

- *The learner makes a prediction which is determined by its policy and the past observations of the process as $B_t := b_t(Z_1, \ldots, Z_{t-1})$.*

---

[1]All subsets of the real numbers are implicitly understood to be endowed with the standard Borel $\sigma$-algebra unless otherwise specified.

- *Nature reveals $Z_t$ to the learner and the learner incurs the loss $\ell(B_t, Z_t)$.*

We will compare the learner's policy to one that is optimal in the sense that it minimizes the expected loss at each time instant under full knowledge of the underlying distribution $P$. That is, an optimal prediction at time instant $t$ is any measurable function $b_t^* : \mathcal{Z}^{t-1} \to \mathcal{B}$ such that

$$b_t^*(z_1, \ldots, z_{t-1}) \in \arg\min_{b \in \mathcal{B}} \mathbb{E}_P\big(\ell(b, Z_t)|Z_1 = z_1, \ldots, Z_{t-1} = z_{t-1}\big), \tag{2}$$

where $\mathbb{E}_P(\cdot|Z_1 = z_1, \ldots, Z_{t-1} = z_{t-1})$ denotes expectation with respect to the measure $K_P^{(t)}(z_1, \ldots, z_{t-1}, \cdot)$. Note that according to (2), for $t = 1$ the function $b_1^*$ takes the tuple $z_1, \ldots, z_0$ as an argument which by convention we regard as an empty tuple and denote as $\emptyset$.

In this section, we study a mismatched prediction problem where the learner is not assumed to know $P$. Instead, we assume that the learner knows[2] some distribution $Q$ which is assumed to be "sufficiently similar"[3] to $P$. For $Q$, we make the same technical assumption regarding disintegration into kernels as we do for $P$. That is, we assume that $Q$ also admits Markov kernels $K_Q^{(t)} : \mathcal{Z}^{t-1} \times \mathcal{G} \to [0, 1]$ so that the analog of (1) holds. In this paper, we study learner's policies that are optimal with regard to $Q$. That is, we study a policy that satisfies, for each $t \in [T]$, that

$$b_t(z_1, \ldots, z_{t-1}) \in \arg\min_{b \in \mathcal{B}} \mathbb{E}_Q\big(\ell(b, Z_t)|Z_1 = z_1, \ldots, Z_{t-1} = z_{t-1}\big). \tag{3}$$

**Remark 1.** *Restricting the analysis to algorithms that can be represented as (3) is less restrictive than it might seem. To illustrate this, we provide detailed arguments in Appendix A that in the following cases, every possible learner's policy has a representation of the form (3).*

1. *$\mathcal{B} = \mathcal{Z}$ is a finite set and the loss function $\ell(b, z) := \mathbb{1}_{b \neq z}$ is the classification loss.*

2. *$\mathcal{Z}$ is finite, $\mathcal{B}$ is the space of all probability mass functions on $\mathcal{Z}$ and the loss function is the self-information loss $\ell(b, z) := -\log(b(z))$.*

The results in this paper hold under the assumption that measurable maps $(b_t^*)_{t \in [T]}$ and $(b_t)_{t \in [T]}$ which satisfy (2) and (3) exist; we implicitly fix suitable choices of these maps. In Appendix B, we show that this is indeed the case as long as mild technical assumptions on $(\mathcal{B}, \mathcal{H})$ and $\ell$ are satisfied.

If the learner executes the policy given in (3), this will yield three random processes embedded in the probability space $(\Omega, \mathcal{F}, \mathbb{P})$: the sequentially observed process $Z_1, \ldots, Z_T$ distributed according to $P$, the sequence $B_1 := b_1(\emptyset), \ldots, B_T := b_T(Z_1, \ldots, Z_{T-1})$ of predictions made by the learner, and the sequence $B_1^* := b_1^*(\emptyset), \ldots, B_T^* := b_T^*(Z_1, \ldots, Z_{T-1})$ of predictions that would have been optimal at each time instant $t \in [T]$ had the learner known $P$. In the following we use $\mathbb{E}$ (without an index) to denote expectation with respect to $\mathbb{P}$.

In this work, we analyze the *regret* that is incurred by the policy given in (3). The *cumulative regret after $t$ rounds* is defined as

$$\Delta_t := \sum_{t'=1}^{t} \big(\ell(B_{t'}, Z_{t'}) - \ell(B_{t'}^*, Z_{t'})\big), \tag{4}$$

and our main quantity of interest is the *average per-round regret*

$$\Delta := \frac{1}{T}\Delta_T. \tag{5}$$

Clearly, if $Q = P$, we can choose $b_t^* = b_t$ for all $t \in [T]$ that satisfy (2) and (3), which yields $\Delta = 0$ almost surely.

We will be considering two main ways of measuring the similarity between two probability distributions $P$ and $Q$. The *variational distance* between $P$ and $Q$ is defined[4] as $\|P - Q\|_{\mathrm{TV}} :=$

---

[2]See Section 6 for a discussion on how the learner can choose a suitable $Q$ even in the absence of specific knowledge about $P$.

[3]The technical notions we use to measure the similarity in this paper are the quantities $V_T$ defined in (8) and $D_T$ defined in (11). For $D_T$, it is worth noting that it is simply a normalized version of the Kullback-Leibler divergence $D(P\|Q)$ via (15) of Lemma 1.

[4]Alternative definitions found in the literature are equivalent to this one, however, sometimes an additional factor of $\frac{1}{2}$ is included in front of the supremum.

$\sup_{F \in \mathscr{G}^{\otimes T}} |P(F) - Q(F)|$. This can, as the notation suggests, be understood as a norm of signed measures, however, for the purposes of this paper it is sufficient to think of it as a metric of probability measures. The *Kullback-Leibler divergence* between $P$ and $Q$ is defined as $D(P||Q) := \mathbb{E}_P \log(dP/dQ)$, where $dP/dQ$ denotes the Radon-Nikodym derivative, and all instances of the functions $\log$ and $\exp$ that appear in this paper are understood with Euler's number as their base. We write $P \ll Q$ if $P$ is absolutely continuous with respect to $Q$, i.e., if every $Q$-null set is also a $P$-null set. If the Radon-Nikodym derivative that appears in the definition of $D(P||Q)$ does not exist because $P \not\ll Q$, we use the convention $D(P||Q) = \infty$.

We define the following notions of similarity of the distributions $P$ and $Q$ which are based on variational distance and Kullback-Leibler divergence and will play a central role in the proofs of our technical results:

$$v_t := \left\| K_P^{(t)}(Z_1, \ldots, Z_{t-1}, \cdot) - K_Q^{(t)}(Z_1, \ldots, Z_{t-1}, \cdot) \right\|_{\mathrm{TV}} \qquad \text{(instantaneous variational distance)} \tag{6}$$

$$\hat{V}_T := \frac{1}{T} \sum_{t=1}^{T} v_t \qquad \text{(average instantaneous variational distance)} \tag{7}$$

$$V_T := \mathbb{E}_P \hat{V}_T, \qquad \text{(expected variational distance)} \tag{8}$$

$$d_t := D\left( K_P^{(t)}(Z_1, \ldots, Z_{t-1}, \cdot) || K_Q^{(t)}(Z_1, \ldots, Z_{t-1}, \cdot) \right) \qquad \text{(instantaneous divergence)} \tag{9}$$

$$\hat{D}_T := \frac{1}{T} \sum_{t=1}^{T} d_t \qquad \text{(average instantaneous divergence)} \tag{10}$$

$$D_T := \mathbb{E}_P \hat{D}_T \qquad \text{(expected divergence)} \tag{11}$$

It is worth noting that the instantaneous notions defined above are random variables while the expected notions are deterministic quantities. This means in particular that instantaneous notions can only be computed given observations of the process $Z_1, \ldots$ while expected notions can be computed without such observations.

## 5 RESULTS FOR THE MISMATCHED PREDICTION PROBLEM

This problem has been solved for *expected regret* by Merhav & Feder (1998). Namely, the authors implicitly prove the following bound:

**Theorem 3** (Merhav & Feder (1998), eq. (23)). *Consider the prediction problem $\mathcal{P}$ defined in Problem 1. The expected average per-round regret $\Delta$ defined in (5) can be bounded as $\mathbb{E}\Delta \leq LV_T$, where $V_T$ is defined in (8).*

This result can be stated in a slightly weaker (but potentially easier to handle) form with the observation that via Pinsker's inequality, the variational distance that appears in Theorem 3 can be upper bounded in terms of Kullback-Leibler divergence which has much nicer tensorization properties. We summarize the relevant connections in the following lemma:

**Lemma 1.** *We have the following relations between the quantities defined in* (6) – (11)*:*

$$v_t \leq \sqrt{\frac{1}{2} d_t} \qquad (12) \qquad\qquad \hat{V}_T \leq \sqrt{\frac{1}{2} \hat{D}_T} \qquad (13) \qquad\qquad V_T \leq \sqrt{\frac{1}{2} D_T}. \qquad (14)$$

*If $P \ll Q$, we also have*

$$D_T = \frac{1}{T} D(P||Q). \tag{15}$$

These (fairly straightforward) facts are implicitly proved[5] for finite $\mathcal{Z}$ by Merhav & Feder (1998). For the sake of completeness, we provide full proofs for Lemma 1 as it is stated here in Appendix C.

In light of (14) and (15) of Lemma 1, we arrive at the following corollary of Theorem 3 which was proposed by Merhav & Feder (1998).

---

[5]Note that the additional factor that appears in Merhav & Feder (1998) is due to the fact that the paper uses a base 2 logarithm in the definition of Kullback-Leibler divergence.

**Corollary 1** (Merhav & Feder (1998), eq. (23)). *Consider the prediction problem $\mathcal{P}$ defined in Problem 1. The expected average per-round regret $\Delta$ defined in (5) can be bounded as $\mathbb{E}\Delta \leq L\sqrt{D(P\|Q)/2T}$.*

In this paper, we study bounds of the same kind as in Theorem 3 and Corollary 1 which hold with high probability over $P$. The main technical result is the following:

**Lemma 2.** *Consider the prediction problem $\mathcal{P}$ defined in Problem 1. For every $\delta \in (0,1]$, with probability at least $1 - \delta$ over $P$, we have*

$$\Delta < 2L\hat{V}_T + \frac{2\sqrt{2}L}{\sqrt{T}} \cdot \sqrt{\log \frac{1}{\delta}}, \tag{16}$$

*where $\hat{V}_T$ is the random variable defined in (7).*

This lemma can be proved by identifying a suitable supermartingale and applying the Azuma-Hoeffding inequality. For full details, we refer the reader to Appendix D.

A problem with the applicability of Lemma 2 is the appearance of the random variable $\hat{V}_T$ in the bound. In order to compute $\hat{V}_T$, we need full knowledge not only of $P$ and $Q$, but also of $Z_1, \ldots, Z_T$. However, we can derive the following high-probability bounds as a consequence of Markov's inequality and the preceding lemmas:

**Theorem 4.** *Consider the prediction problem $\mathcal{P}$ defined in Problem 1. For every $\delta \in (0,1]$, each of the following inequalities holds with probability at least $1 - \delta$ over $P$ (where $V_T$ is given in (8)):*

$$\Delta < 4LV_T \cdot \frac{1}{\delta} + \frac{2\sqrt{2}L}{\sqrt{T}}\sqrt{\log \frac{2}{\delta}} \quad (17) \quad \Delta < 2L\sqrt{\frac{D(P\|Q)}{T}} \cdot \frac{1}{\sqrt{\delta}} + \frac{2\sqrt{2}L}{\sqrt{T}}\sqrt{\log \frac{2}{\delta}}. \quad (18)$$

We include the full technical details of the proof in Appendix E. The usefulness of Theorem 4 is not immediately clear from its statement since computation of the bounds requires knowledge of $P$ and $Q$ and the learner is in general not expected to know $P$. However, these quantities can conceivably be upper bounded in case the learner has knowledge that $Q$ is a sufficiently high-quality approximation of $P$. One case which is particularly relevant and for which it is known how to construct a suitable $Q$ alongside a bound for $D(P\|Q)$ is the universal prediction scenario. We describe this scenario, survey the literature on how to choose $Q$, bound $D(P\|Q)$, and obtain a high-probability regret bound from (18) in Section 6.

Since these concentration results are obtained from Lemma 2 with the Markov bound (which is known not to be tight in many cases), this begs the question of whether it would be possible to improve upon Theorem 4 by using more advanced techniques. It is not unheard of that learning algorithms perform well in expectation but in general pick up factors that are polynomial in the tail probability when viewed through the high-probability lens. This was for instance shown by Aden-Ali et al. (2023) for a class of binary classification problems. With the following impossibility result, we show that this is also the case here and this aspect of the bound cannot be improved without making additional assumptions.

**Theorem 5.** *For every $C \in (0,\infty)$, $\alpha \in [0,1)$, $\beta \in [0,1/2)$, and every $\varepsilon : \mathbb{N} \times (0,1] \to [0,\infty)$ such that for every fixed $\delta \in (0,1]$, $\varepsilon(T,\delta) \to 0$ as $T \to \infty$, there is a prediction problem $\mathcal{P} = (T, (\mathcal{Z}, \mathcal{G}), P, (\mathcal{B}, \mathcal{H}), \ell)$ with the following properties: (i) $\mathcal{Z}, \mathcal{B} := \{0,1,2\}$ (with discrete $\sigma$-algebras), and $\ell(b,z) = \mathbb{1}_{b \neq z}$ is the classification loss; (ii) there exist a probability distribution $Q$ on $(\mathcal{Z}^T, \mathcal{G}^{\otimes T})$ and $\delta \in (0,1)$ as well as a learner's policy satisfying (3) such that with probability at least $\delta$ under $P$, the average per-round regret $\Delta$ defined in (5) satisfies the following lower bounds (where $V_T$ is defined in (8)):*

$$\Delta \geq C \cdot V_T \cdot \frac{1}{\delta^{\alpha}} + \varepsilon(T,\delta) \quad (19) \qquad \Delta \geq C \cdot \sqrt{\frac{D(P\|Q)}{T}} \cdot \frac{1}{\delta^{\beta}} + \varepsilon(T,\delta). \quad (20)$$

To prove Theorem 5, we construct a prediction problem in which the regret is $0$ in most cases, but there is a non-negligible probability that the average per-round regret is close to $1$ (which is the maximum value for the classification loss function given in the theorem statement). We then

argue that because we have this non-negligible probability of high regret, it is not possible to give a high-probability bound that converges to a value significantly below 1. The full details of this construction are relegated to the proof in Appendix F.

We can see from Theorem 5 that Theorem 4 cannot be improved in the sense that the factor $1/\delta$, respectively $1/\sqrt{\delta}$, that appears in the bounds cannot be replaced with an expression that grows significantly more slowly (order-wise) as $\delta \to 0$ without making additional assumptions. This remains true even if we are willing to sacrifice convergence speed of the second summand that appears in the bound.

# 6    APPLYING MISMATCHED PREDICTION RESULTS TO UNIVERSAL PREDICTION

In this section, we discuss the relevance of the results proposed in Section 5 and how they could be applied to online learning problems. The most obvious way our results can be used is if a good approximation $Q$ of $P$ is known, in which case, as long as we can bound $D(P||Q)$ or $V_T$, Theorem 4 is immediately applicable. In this section, we consider the case that a suitable $Q$ is not available to the learner a priori, focusing on the rate at which $\Delta$ converges to 0 as $T \to \infty$. To formulate this precisely, we need the following asymptotic notion of the problem of prediction of stochastic sequences:

**Problem 2** (Asymptotic prediction of stochastic sequences). *An asymptotic prediction problem $\mathcal{A} = ((\mathcal{Z}, \mathcal{G}), P_{\mathbb{N}}, (\mathcal{B}, \mathcal{H}), \ell)$ is characterized by a measurable space $(\mathcal{Z}, \mathcal{G})$, a probability distribution $P_{\mathbb{N}}$ on $(\mathcal{Z}^{\mathbb{N}}, \mathcal{G}^{\otimes \mathbb{N}})$, a measurable space $(\mathcal{B}, \mathcal{H})$ (the prediction domain), and a measurable loss function $\ell : \mathcal{B} \times \mathcal{Z} \to [0, L]$, where $L \in [0, \infty)$. The learner's policy $b = (b_t)_{t \in \mathbb{N}}$ consists of measurable (deterministic) functions $b_t : \mathcal{Z}^{t-1} \to \mathcal{B}$.*

For each $T \in \mathbb{N}$, an asymptotic prediction problem $\mathcal{A} = ((\mathcal{Z}, \mathcal{G}), P_{\mathbb{N}}, (\mathcal{B}, \mathcal{H}), \ell)$ induces a prediction problem (as defined in Problem 1) $\mathcal{P}_T := (T, (\mathcal{Z}, \mathcal{G}), P_{[T]}, (\mathcal{B}, \mathcal{H}), \ell)$ and a learner's policy $(b_t)_{t \in \mathbb{N}}$ for $\mathcal{A}$ induces a policy $(b_t)_{t \in [T]}$ for $\mathcal{P}_T$.

We consider a learner's policy that satisfies, similarly to (3),

$$b_t(z_1, \ldots, z_{t-1}) \in \underset{b \in \mathcal{B}}{\arg\min} \, \mathbb{E}_{Q_{\mathbb{N}}}\big(\ell(b, Z_t)|Z_1 = z_1, \ldots, Z_{t-1} = z_{t-1}\big). \tag{21}$$

If $T$ is clear from context, we will sometimes write $P$ instead of $P_{[T]}$ and $Q$ instead of $Q_{[T]}$ to match notation in the rest of the paper. Because no random variables other than $Z_1, \ldots, Z_t$ appear in (21) and $t \leq T$, any policy function $b_t$ satisfies (21) iff it satisfies (3), making the results in Section 5 applicable.

We are interested in whether the average per-round regret $\Delta$ defined in (5) converges to 0 as $T \to \infty$ and if it does, what the convergence order is. It is clear from Corollary 1 and Theorem 4 that whenever $D(P||Q)$ grows sublinearly in $T$, the average per-round regret $\Delta$ converges to 0 both in expectation and with high probability.

The scenario usually considered in the literature is the one proposed by Merhav & Feder (1998) where there is some parametrized family $(P_\theta)_{\theta \in \Theta}$ (where $\Theta$ is a measurable space) that is known to the learner and it is further known that $P_{\mathbb{N}} = P_{\theta_0}$ for some $\theta_0 \in \Theta$ (where $\theta_0$ does not need to be known by the learner). Under this assumption, it was proposed by Merhav & Feder (1998) that the learner chooses a learner's policy which satisfies (21) with respect to the *universal measure*

$$Q_{\mathbb{N}} := \int_\Theta P_\theta w(d\theta), \tag{22}$$

where $w$ is a probability distribution on $\Theta$.

There are two main cases we are aware of in which (22) is well-defined and yields a $Q_{\mathbb{N}}$ which describes a policy that has a regret which vanishes at a known rate: (i) $\Theta$ is countable, or (ii) $(P_\theta)_{\theta \in \Theta}$ satisfies suitable smoothness conditions and all $P_\theta$ are i.i.d. or Markov chains with finite memory.

In case (i), it is observed by Hutter (2003) that $D(P||Q) \leq -\log w(\{\theta_0\})$, independently of $T$. Therefore, as long as $w(\{\theta\}) > 0$ for all $\theta \in \Theta$ (which is always possible for countable $\Theta$), our Theorem 4 yields an $\mathcal{O}(T^{-1/2}\delta^{-1/2})$ convergence rate with probability at least $1 - \delta$ compared to

| | In expectation (Merhav & Feder, 1998) | With probability $\geq 1 - \delta$ (this work) |
|---|---|---|
| $\mathcal{Z}$ countable (Hutter, 2003) | $\mathcal{O}(T^{-1/2})$ | $\mathcal{O}(T)^{-1/2}\delta^{-1/2})$ |
| $P_{\mathbb{N}}$ i.i.d. (Clarke & Barron, 1990) | $\mathcal{O}((T/\log T)^{-1/2})$ | $\mathcal{O}((T/\log T)^{-1/2}\delta^{-1/2})$ |
| $P_{\mathbb{N}}$ Markov with memory (Atteson, 1999) | $\mathcal{O}((T/\log T)^{-1/2})$ | $\mathcal{O}(T/\log T)^{-1/2}\delta^{-1/2})$ |

Table 1: Summary of the convergence behavior of the average per-round regret $\Delta$ for different assumptions about $P_{\mathbb{N}}$, respectively $\mathcal{Z}$ in the case where $P_{\mathbb{N}}$ comes from a parametrized family.

the $\mathcal{O}(T^{-1/2})$ convergence rate of the expected regret guaranteed by Corollary 1. It was further noted by Hutter (2003) that the learner's knowledge of a suitable parametrized family $(P_\theta)_{\theta \in \Theta}$ is actually not a very restrictive assumption at all: it is possible to enumerate all probability distributions on $\mathcal{Z}^{\mathbb{N}}$ that are computable to arbitrary precision on a universal Turing machine. Although this has no impact on the convergence rate, the constant that appears in the regret bound could be extremely large if this is done. Therefore, Hutter (2003) suggested to assign higher probabilities in $w$ to distributions with lower Kolmogorov complexity. In this way, as long as the distribution $P_{\mathbb{N}}$ is of sufficiently low Kolmogorov complexity, the constant in the regret bound would not be overly large.

Next, we discuss case (ii). Merhav & Feder (1998) point out that Clarke & Barron (1990) show for the case in which $P_{\mathbb{N}}$ is an i.i.d. distribution that the distribution $Q_{\mathbb{N}}$ defined in (22) has the property that $D(P||Q)$ grows as $\mathcal{O}(\log(T))$. This holds if $w$ is supported on all of $\Theta$ and some additional smoothness assumptions are satisfied. Consequently, $\Delta$ converges as $\mathcal{O}((T/\log T)^{-1/2})$ in expectation by Corollary 1 and our Theorem 4 yields a convergence order of $\mathcal{O}((T/\log T)^{-1/2}\delta^{-1/2})$ with probability at least $1 - \delta$. It was later shown by Atteson (1999) that the same holds not just for the i.i.d. case, but also for the more general case of Markov chains with memory of a fixed order.

The convergence rates of $\Delta$ are summarized in Table 1. In multiple relevant cases, Corollary 1 and Theorem 4 yield vanishing average per-round regret and an upper bound on the convergence order. This is assuming that $P_{\mathbb{N}}$ comes from a known parametrized family $(P_\theta)_{\theta \in \Theta}$, but as noted by Hutter (2003), often this is not an overly limiting assumption.

## 7 NUMERICAL EXPERIMENTS

In this section, we apply the algorithm (21) with a choice of $Q_{\mathbb{N}}$ of the form given in (22) to a Markov chain with memory and a finite state space. Specifically, $\mathcal{Z} = \{1, \ldots, S\}$ where $S$ denotes the number of states the Markov chain can take. The probability distribution $P_{\mathbb{N}}$ of the underlying process $Z_1, \ldots$ is then the unique distribution that satisfies (1) for all $t \in \mathbb{N}$ with

$$K_P^{(t)}(z_1, \ldots, z_{t-1}, \{z_t\}) := p_{z_t | z_{t-m}, \ldots, z_{t-1}}$$

for all $z_1, \ldots, z_t \in \mathcal{Z}$, where for simplicity we use the convention that $z_t = 1$ whenever $t \leq 0$ (i.e., the initial few realizations of the process are as though the missing realizations were all in state 1) and $(p_{s|s_1, \ldots, s_m})_{s, s_1, \ldots, s_m \in \mathcal{Z}}$ are parameters with the property

$$\forall s, s_1, \ldots, s_m \in \mathcal{Z} : p_{s|s_1, \ldots, s_m} \geq 0, \quad \forall s_1, \ldots, s_m \in \mathcal{Z} : \sum_{s=1}^{S} p_{s|s_1, \ldots, s_m} = 1. \quad (23)$$

The learner's prediction task is to predict the next realization of the Markov chain, i.e., $\mathcal{B} = \mathcal{Z}$, and its performance is evaluated with the classification loss, $\ell(b, z) := \mathbb{1}_{b \neq z}$.

The parameters in (23) are unknown to the learner, which will instead rely on $Q_{\mathbb{N}}$ defined in (22), where $\Theta$ is the set of all possible tuples of $(p_{s|s_1, \ldots, s_m})_{s, s_1, \ldots, s_m \in \mathcal{Z}}$ which satisfy (23). We relegate the details of how $\Theta$ is constructed and how the expectation over the resulting probability distribution $Q$ is minimized to Appendix H.

In Figure 1, we show the mean regret of the learner's policy as well as the observed regret quantiles in $4,000$ simulation runs of this policy. It can be seen that the average regret gets close to 0 for long enough sequence lengths, and so do the quantiles (which can be understood as empirical observations

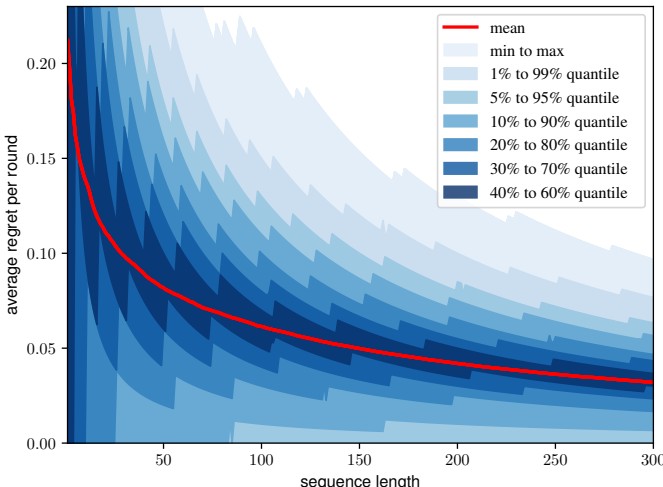

Figure 1: Average regret per round for a Markov chain with memory order $m = 3$ and $S = 2$ states. Shown are the mean and selected quantiles for $4,000$ runs.

of the high-probability regret), albeit quite a bit more slowly than the average, as would be expected from our theoretical results.

In this example, we have chosen to enumerate *all* possible probability distributions for $m = 3$ and $S = 2$ in our family $\Theta$. As can be seen in the definition (47) of $\Theta$ in Appendix H, the dimension of $\Theta$ is $(S - 1) \cdot S^m$ which remains manageable as long as the memory order is small and the state space moderately sized. However, the framework also offers the flexibility to incorporate additional side information about the distribution of the observed sequence by choosing to include only certain probability distributions in $\Theta$. In this way, the convergence time can be improved, computing the predictions can be made more efficient, and using larger (or continuous) state spaces can become tractable. If the hyperparameters of the Metropolis-Hastings algorithm are appropriately tuned, we expect roughly a linear dependence of the computational complexity on the dimension of $\Theta$. As a rough orientation for the complexity of this example, it takes between a few seconds and a few minutes to generate a single prediction on a standard laptop without parallelization (depending on the length of the already observed sequence). It it also worth noting that $w$ does *not* need to accurately represent prior belief to guarantee a vanishing regret – indeed, the convergence rates in Table 1 are guaranteed to hold as long as the true $\theta$ is in the support of $w$. With that being said, it is possible to incorporate prior belief into the design of $w$ by assigning more weight to more likely $\theta$. While that will not affect the convergence rate, it does affect the constants that are hidden in the $\mathcal{O}$ expressions and can therefore be attractive in practice.

## 8 LIMITATIONS AND FUTURE RESEARCH DIRECTIONS

While the mismatched prediction results of Theorem 4 need very few assumptions and can therefore be expected to be immediately applicable to a wide range of practical problems, their usefulness for universal prediction scenarios would crucially depend on the construction of a suitable parametric family $(P_\theta)_{\theta \in \Theta}$. The literature we summarize in Section 6 has several examples of how to do this that are quite satisfying from a mathematical perspective, and even the idea of enumerating all possible probability distributions can sometimes be feasible as we show in Section 7. However, this might not be feasible in every practical scenarios, so additional research might be needed for specific scenarios to find suitable approximations of the minimization (3) when the parametric family is very large.

It is also interesting that the error probability $\delta$ in the upper bound of Lemma 2 appears only in the form $\sqrt{\log(1/\delta)}$, but the scaling behavior of the error terms in Theorem 4 is much worse. We show in Theorem 5 that it is not possible to derive a better result in general, but it would be worthwhile to investigate in future research if there are suitable additional assumptions that would yield a bound akin to the one in Theorem 4 but with a logarithmic scaling in $\delta$ comparable to the bound of Lemma 2.

ACKNOWLEDGMENTS

This work was supported in part by the Australian Research Council under Project DP230101493.

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

## A   DETAILED ARGUMENTS FOR REMARK 1

**Item 1:** Consider a learner's policy which for all $t \in [T]$, having seen realizations $z_1, \ldots, z_{t-1}$ makes the prediction $b_t(z_1, \ldots, z_{t-1})$. We design $Q$ such that conditioned under $z_1, \ldots, z_{t-1}$, it assigns a large probability to $b_t(z_1, \ldots, z_{t-1})$ and smaller but nonzero probabilities (so that no singularities in the definition of conditional probability occur) to the other possible outcomes. More precisely, pick $q, r \in (0, 1)$ such that $q > r/(|\mathcal{Z}| - 1)$ and $q + r = 1$. Define, for each $t$,

$$K_Q^{(t)}(z_1, \ldots, z_{t-1}, \{z\}) := \begin{cases} q, & z = b_t(z_1, \ldots, z_{t-1}) \\ r/(|\mathcal{Z}| - 1), & \text{otherwise,} \end{cases}$$

and construct $Q$ such that it satisfies the analog of (1). Then the policy $(b_t)_{t \in [T]}$, which was arbitrary above, satisfies (3).

**Item 2:** Given a learner's policy $(b_t)_{t \in [T]}$, we can define $K_Q^{(t)}(z_1, \ldots, z_{t-1}, \{z\}) := b_t(z_1, \ldots, z_{t-1})(z)$. We then have

$$\mathbb{E}_Q\big(\ell(b, Z_t)|Z_1 = z_1, \ldots, Z_{t-1} = z_{t-1}\big) = -\mathbb{E}_{b_t(z_1, \ldots, z_{t-1})} \log(b(Z_t))$$
$$= H(b_t(z_1, \ldots, z_{t-1})) + D\left(b_t(z_1, \ldots, z_{t-1}) \| b\right),$$

where $D\left(\cdot \| \cdot\right)$ denotes Kullback-Leibler divergence and $H$ denotes Shannon entropy. Since Kullback-Leibler divergence takes its minimum value 0 iff the two arguments are identical, we can conclude that this expression is minimized for $b = b_t(z_1, \ldots, z_{t-1})$, showing that the policy is indeed represented in the form (3).

## B   EXISTENCE OF MEASURABLE $(b_t^*)_{t \in [T]}$ AND $(b_t)_{t \in [T]}$

In this appendix, we give sufficient conditions under which there exist measurable maps $(b_t^*)_{t \in [T]}$ and $(b_t)_{t \in [T]}$ such that (2) and (3) are satisfied.

The measurable maximum theorem (Aliprantis & Border, 2006, Theorem 18.19) ensures that such maps exist if $(\mathcal{B}, \mathcal{H})$ is metrizable in such a way that the following are satisfied:

- $(\mathcal{B}, \mathcal{H})$ is separable and compact.
- For every $b \in \mathcal{B}$, the function $\ell(b, \cdot) : \mathcal{Z} \to [0, L]$ is measurable.
- The function $\ell(\cdot, z) : \mathcal{B} \to [0, L]$ is continuous.

An important special case of this is that of finite $\mathcal{B}$ with the discrete $\sigma$-algebra as the choice for $\mathcal{H}$.

## C   PROOF OF LEMMA 1

First, we observe that (12) is simply an application of Pinsker's inequality.

For (13), we argue

$$\hat{V}_T \overset{(7)}{=} \frac{1}{T} \sum_{t=1}^{T} v_t \overset{(12)}{\leq} \frac{1}{T} \sum_{t=1}^{T} \sqrt{\frac{1}{2} d_t} \overset{(a)}{\leq} \sqrt{\frac{1}{2} \cdot \frac{1}{T} \sum_{t=1}^{T} d_t} \overset{(10)}{=} \sqrt{\frac{1}{2} \hat{D}_T},$$

where (a) is due to Jensen's inequality and concavity of the square root.

For (14), we make the very similar argument

$$V_T \overset{(8)}{=} \mathbb{E}_P \hat{V}_T \overset{(13)}{\leq} \mathbb{E}_P \sqrt{\frac{1}{2} \hat{D}_T} \overset{(a)}{\leq} \sqrt{\frac{1}{2} \mathbb{E}_P \hat{D}_T} \overset{(11)}{=} \sqrt{\frac{1}{2} D_T},$$

where (a) is again due to Jensen's inequality and concavity of the square root.

For (15), we first note that since $P \ll Q$, it follows by induction from (Rao, 2004, Section 6.3, Proposition 1) that $P$-almost surely for all $t \in [T]$, $K_P^{(t)}(Z_1, \ldots, Z_{t-1}, \cdot) \ll K_Q^{(t)}(Z_1, \ldots, Z_{t-1}, \cdot)$, and we can decompose, for $P$-almost all $(z_1, \ldots, z_T) \in \mathcal{Z}^T$,

$$\frac{dP}{dQ}(z_1, \ldots, z_T) = \prod_{t=1}^{T} \frac{dK_P^{(t)}(z_1, \ldots, z_{t-1}, \cdot)}{dK_Q^{(t)}(z_1, \ldots, z_{t-1}, \cdot)}(z_t). \tag{24}$$

So we have

$$\frac{1}{T} D\left(P \| Q\right) \stackrel{(a)}{=} \frac{1}{T} \mathbb{E}_P \log \frac{dP}{dQ}(Z_1, \ldots, Z_T)$$

$$\stackrel{(24)}{=} \frac{1}{T} \mathbb{E}_P \log \prod_{t=1}^{T} \frac{dK_P^{(t)}(Z_1, \ldots, Z_{t-1}, \cdot)}{dK_Q^{(t)}(Z_1, \ldots, Z_{t-1}, \cdot)}(Z_t)$$

$$\stackrel{(b)}{=} \sum_{t=1}^{T} \frac{1}{T} \mathbb{E}_{P_{[t-1]}} \mathbb{E}_{K_P^{(t)}(Z_1, \ldots, Z_{t-1}, \cdot)} \log \frac{dK_P^{(t)}(Z_1, \ldots, Z_{t-1}, \cdot)}{dK_Q^{(t)}(Z_1, \ldots, Z_{t-1}, \cdot)}(Z_t)$$

$$\stackrel{(a)}{=} \sum_{t=1}^{T} \frac{1}{T} \mathbb{E}_{P_{[t-1]}} d_t$$

$$\stackrel{(c)}{=} \mathbb{E} \frac{1}{T} \sum_{t=1}^{T} d_t$$

$$\stackrel{(10),(11)}{=} D_T,$$

where in both steps labeled (a) we have used the definition of Kullback-Leibler divergence, in (b) we have used (1) in conjunction with the linearity of expectation, and in (d) we have used the linearity of expectation.

## D    PROOF OF LEMMA 2

Property (3) of $b_t(z_1, \ldots, z_{t-1})$ implies in particular that for every $t \in [T]$ and $z_1, \ldots, z_{t-1}$,

$$\mathbb{E}_Q\left(\ell(b_t(z_1, \ldots, z_{t-1}), Z_t)|Z_1 = z_1, \ldots, Z_{t-1} = z_{t-1}\right)$$
$$\leq \mathbb{E}_Q\left(\ell(b_t^*(z_1, \ldots, z_{t-1}), Z_t)|Z_1 = z_1, \ldots, Z_{t-1} = z_{t-1}\right). \tag{25}$$

We can further bound the left-hand side as

$$\mathbb{E}_Q\left(\ell(b_t(z_1, \ldots, z_{t-1}), Z_t)|Z_1 = z_1, \ldots, Z_{t-1} = z_{t-1}\right)$$

$$= \int_0^{\infty} K_Q^{(t)}\left(z_1, \ldots, z_{t-1}, \{z_t \in \mathcal{Z} : \ell(b_t(z_1, \ldots, z_{t-1}), z_t) > x\}\right) dx$$

$$\stackrel{(a)}{=} \int_0^L K_Q^{(t)}\left(z_1, \ldots, z_{t-1}, \{z_t \in \mathcal{Z} : \ell(b_t(z_1, \ldots, z_{t-1}), z_t) > x\}\right) dx$$

$$\geq \int_0^L \left( K_P^{(t)}\left(z_1, \ldots, z_{t-1}, \{z_t \in \mathcal{Z} : \ell(b_t(z_1, \ldots, z_{t-1}), z_t) > x\}\right) \right.$$

$$\left. - \left\| K_P^{(t)}(z_1, \ldots, z_{t-1}, \cdot) - K_Q^{(t)}(z_1, \ldots, z_{t-1}, \cdot) \right\|_{\text{TV}} \right) dx$$

$$= \mathbb{E}_P\left(\ell(b_t(z_1, \ldots, z_{t-1}), Z_t)|Z_1 = z_1, \ldots, Z_{t-1} = z_{t-1}\right) \tag{26}$$

$$- L \left\| K_P^{(t)}(z_1, \ldots, z_{t-1}, \cdot) - K_Q^{(t)}(z_1, \ldots, z_{t-1}, \cdot) \right\|_{\text{TV}},$$

where (a) is due to the loss being bounded in $[0, L]$. With the same argument, we can bound the right-hand side of (25) as

$$\mathbb{E}_Q\big(\ell(b_t^*(z_1, \ldots, z_{t-1}), Z_t)|Z_1 = z_1, \ldots, Z_{t-1} = z_{t-1}\big)$$
$$\leq \mathbb{E}_P\big(\ell(b_t^*(z_1, \ldots, z_{t-1}), Z_t)|Z_1 = z_1, \ldots, Z_{t-1} = z_{t-1}\big)$$
$$+ L \left\| K_P^{(t)}(z_1, \ldots, z_{t-1}, \cdot) - K_Q^{(t)}(z_1, \ldots, z_{t-1}, \cdot) \right\|_{\text{TV}}. \quad (27)$$

Combining (25), (26), and (27), we obtain

$$\mathbb{E}_P\big(\ell(b_t(z_1, \ldots, z_{t-1}), Z_t)|Z_1 = z_1, \ldots, Z_{t-1} = z_{t-1}\big)$$
$$- \mathbb{E}_P\big(\ell(b_t^*(z_1, \ldots, z_{t-1}), Z_t)|Z_1 = z_1, \ldots, Z_{t-1} = z_{t-1}\big)$$
$$- 2L \left\| K_P^{(t)}(z_1, \ldots, z_{t-1}, \cdot) - K_Q^{(t)}(z_1, \ldots, z_{t-1}, \cdot) \right\|_{\text{TV}} \leq 0. \quad (28)$$

Note that if we define

$$\mathbb{E}\Big(\ell(B_t, Z_t)|Z_1, \ldots, Z_{t-1}\Big)(\omega)$$
$$:= \int_{\mathcal{Z}} \ell\Big(b_t\big(Z_1(\omega), \ldots, Z_{t-1}(\omega)\big), z_t\Big) K_P^{(t)}(Z_1(\omega), \ldots, Z_{t-1}(\omega), dz_t),$$

then $\mathbb{E}\Big(\ell(B_t, Z_t)|Z_1, \ldots, Z_{t-1}\Big)$ is (as the notation suggests) a version of the conditional expectation due to (Çinlar, 2011, Section IV, Theorem 2.19). Clearly, the analog for the conditional expectation $\mathbb{E}\Big(\ell(B_t^*, Z_t)|Z_1, \ldots, Z_{t-1}\Big)$ holds as well. Also noting (6), we can therefore rewrite (28) as

$$\mathbb{E}\Big(\ell(B_t, Z_t)|Z_1, \ldots, Z_{t-1}\Big) - \mathbb{E}\Big(\ell(B_t^*, Z_t)|Z_1, \ldots, Z_{t-1}\Big) - 2Lv_t \leq 0 \quad (29)$$

almost surely. Next, we define an auxiliary process $\Delta_0', \ldots, \Delta_T'$ via $\Delta_0' := 0$ and

$$\Delta_t' := \sum_{t'=1}^t \Big(\ell(B_{t'}, Z_{t'}) - \ell(B_{t'}^*, Z_{t'}) - 2Lv_{t'}\Big)$$

for $t \in [T]$. We calculate

$$\mathbb{E}\big(\Delta_t'|Z_1, \ldots, Z_{t-1}\big)$$
$$= \mathbb{E}\Big(\Delta_{t-1}' + \ell(B_t, Z_t) - \ell(B_t^*, Z_t) - 2Lv_t|Z_1, \ldots, Z_{t-1}\Big)$$
$$\overset{(a)}{=} \Delta_{t-1}' + \mathbb{E}\Big(\ell(B_t, Z_t)|Z_1, \ldots, Z_{t-1}\Big) - \mathbb{E}\Big(\ell(B_t^*, Z_t)|Z_1, \ldots, Z_{t-1}\Big) - 2Lv_t$$
$$\overset{(29)}{\leq} \Delta_{t-1}'$$

where in step (a) we have used the linearity of conditional expectation and the fact that $\Delta_{t-1}'$ as well as the variational distance term are measurable in $\sigma(Z_1, \ldots, Z_{t-1})$. It follows that $\Delta_0', \ldots, \Delta_T'$ is a supermartingale. Next, we observe that for every $t \in [T]$, we have

$$\Delta_t' - \Delta_{t-1}' = \ell(B_t, Z_t) - \ell(B_t^*, Z_t) - 2Lv_t \in [-3L, L].$$

An application of the version of the Azuma-Hoeffding inequality stated in Theorem 6 of Appendix G yields, for any $\varepsilon \in (0, \infty)$,

$$P\big(\Delta_T' \geq \varepsilon\big) \leq \exp\left(-\frac{2\varepsilon^2}{T \cdot (4L)^2}\right) = \exp\left(-\frac{\varepsilon^2}{8TL^2}\right).$$

With the definition

$$\varepsilon := 2\sqrt{2T}L\sqrt{\log \frac{1}{\delta}},$$

which can be equivalently written as

$$\delta = \exp\left(-\frac{\varepsilon^2}{8TL^2}\right),$$

we obtain that with probability at least $1 - \delta$ over $P$, we have

$$\Delta'_T < 2\sqrt{2T}L\sqrt{\log\frac{1}{\delta}}. \tag{30}$$

To conclude the proof of the lemma, we observe that

$$\Delta'_T = T\Delta - 2L\sum_{t=1}^{T} v_t \stackrel{(7)}{=} T\left(\Delta - 2L\hat{V}_T\right)$$

which we substitute into (30) and solve for $\Delta$.

## E    PROOF OF THEOREM 4

**Lemma 3.** *Consider the prediction problem $\mathcal{P}$ defined in Problem 1 and the quantities defined in (6) – (11). For every $\delta \in (0, 1]$, each of the following holds with probability at least $1 - \delta$ over $P$:*

$$\hat{V}_T < V_T \cdot \frac{1}{\delta} \tag{31}$$

$$\hat{V}_T < \sqrt{\frac{1}{2\delta}D_T}. \tag{32}$$

*Proof.* (31) is a direct consequence of the definition of $V_T$ in (8) and Markov's inequality. For (32), we use Lemma 1 alongside definitions (7) – (11) and Markov's inequality. Specifically, we observe that due to Markov's inequality, we have

$$P\left(\hat{D}_T \geq \frac{1}{\delta}D_T\right) \leq \frac{\mathbb{E}\hat{D}_T}{\frac{1}{\delta}D_T} \stackrel{(11)}{=} \delta. \tag{33}$$

With this, we can argue that with probability at least $1 - \delta$ over $P$,

$$\hat{V}_T \stackrel{(7)}{=} \frac{1}{T}\sum_{t=1}^{T} v_t \stackrel{(12)}{\leq} \frac{1}{T}\sum_{t=1}^{T}\sqrt{\frac{1}{2}d_t} \stackrel{(a)}{\leq} \sqrt{\frac{1}{T}\sum_{t=1}^{T}\frac{1}{2}d_t} \stackrel{(10)}{=} \sqrt{\frac{1}{2}\hat{D}_T} \stackrel{(33)}{<} \sqrt{\frac{1}{2\delta}D_T},$$

where step (a) is due to Jensen's inequality and concavity of the square root. $\qquad\square$

*Proof of Theorem 4.* We first observe that the implication

$$\Delta < 2L\hat{V}_T + \frac{2\sqrt{2}L}{\sqrt{T}}\sqrt{\log\frac{2}{\delta}} \wedge 2L\hat{V}_T < 2LV_T \cdot \frac{2}{\delta} \Rightarrow \Delta < 2LV_T \cdot \frac{2}{\delta} + \frac{2\sqrt{2}L}{\sqrt{T}}\sqrt{\log\frac{2}{\delta}} \tag{34}$$

holds. With this, we argue

$$P\left(\Delta \geq 2LV_T \cdot \frac{2}{\delta} + \frac{2\sqrt{2}L}{\sqrt{T}}\sqrt{\log\frac{2}{\delta}}\right)$$

$$\stackrel{(a)}{\leq} P\left(\Delta \geq 2L\hat{V}_T + \frac{2\sqrt{2}L}{\sqrt{T}}\sqrt{\log\frac{2}{\delta}} \vee 2L\hat{V}_T \geq 2LV_T \cdot \frac{2}{\delta}\right)$$

$$\stackrel{(b)}{\leq} P\left(\Delta \geq 2L\hat{V}_T + \frac{2\sqrt{2}L}{\sqrt{T}}\sqrt{\log\frac{2}{\delta}}\right) + P\left(2L\hat{V}_T \geq 2LV_T \cdot \frac{2}{\delta}\right)$$

$$\stackrel{(c)}{\leq} \frac{\delta}{2} + \frac{\delta}{2},$$

where (a) is by monotonicity of probability using the contrapositive of (34), (b) is due to the union bound, and in (c) we have used Lemma 2 the first summand and (31) of Lemma 3 in the second summand, where $\delta/2$ has been substituted for $\delta$ in both cases.

For (18), we first note that if $P \not\ll Q$, the bound trivially holds since the right-hand side is infinite in this case. For the case $P \ll Q$, the proof proceeds along very similar lines as before. Namely, we observe that the implication

$$\Delta < 2L\hat{V}_T + \frac{2\sqrt{2}L}{\sqrt{T}} \cdot \sqrt{\log \frac{2}{\delta}} \wedge \hat{V}_T < \sqrt{\frac{1}{T\delta}D(P\|Q)}$$

$$\Rightarrow \Delta < 2L\sqrt{\frac{1}{T\delta}D(P\|Q)} + \frac{2\sqrt{2}L}{\sqrt{T}} \cdot \sqrt{\log \frac{2}{\delta}}. \quad (35)$$

holds. Hence, we can argue, for every $\delta \in (0, 1]$,

$$P\left(\Delta \geq 2L\sqrt{\frac{D(P\|Q)}{T}} \cdot \frac{1}{\sqrt{\delta}} + \frac{2\sqrt{2}L}{\sqrt{T}}\sqrt{\log \frac{2}{\delta}}\right)$$

$$\overset{(a)}{\leq} P\left(\Delta \geq 2L\hat{V}_T + \frac{2\sqrt{2}L}{\sqrt{T}} \cdot \sqrt{\log \frac{2}{\delta}} \vee \hat{V}_T \geq \sqrt{\frac{1}{T\delta}D(P\|Q)}\right)$$

$$\overset{(b)}{\leq} P\left(\Delta \geq 2L\hat{V}_T + \frac{2\sqrt{2}L}{\sqrt{T}} \cdot \sqrt{\log \frac{2}{\delta}}\right) + P\left(\hat{V}_T \geq \sqrt{\frac{1}{T\delta}D(P\|Q)}\right)$$

$$\overset{(15)}{=} P\left(\Delta \geq 2L\hat{V}_T + \frac{2\sqrt{2}L}{\sqrt{T}} \cdot \sqrt{\log \frac{2}{\delta}}\right) + P\left(\hat{V}_T \geq \sqrt{\frac{1}{\delta}D_T}\right)$$

$$\overset{(c)}{\leq} \frac{\delta}{2} + \frac{\delta}{2},$$

where in (a) we have used the contrapositive of (35) with the monotonicity of probability, in (b) we have used the union bound, and in (c) we have used Lemma 2 in the first summand and (32) of Lemma 3 in the second summand, where $\delta/2$ is substituted for $\delta$ in both cases. $\qquad\square$

## F  PROOF OF THEOREM 5

This proof is structured as follows: we first construct two probability distributions $P_\mathbb{N}$ and $Q_\mathbb{N}$ on $\mathcal{Z}^\mathbb{N}$ with two free parameters $\varphi$ and $\psi$. This is done with the understanding that once $T$ is fixed, $P := P_{[T]}$ and $Q := Q_{[T]}$ are the marginals on the first $T$ components. We then proceed to calculating $\hat{V}_T$ and $D(P\|Q)$ in dependence of $\varphi, \psi$, and $T$. Next, we determine predictors $B_1, \ldots, B_T$ that are associated with a learner's policy that satisfies (3), as well as the predictors $B_1^*, \ldots, B_T^*$ associated with an optimal prediction rule that satisfies (2) (which also exists since $\mathcal{B}$ is finite and $\mathcal{H}$ discrete). Based on this, we determine the average per-round regret $\Delta$. Finally, we specialize this general construction to specific choices of $\varphi, \psi, T$, and $\delta$ depending on $C, \alpha, \beta$, and $\varepsilon$ that satisfy (19) and (20).

**Choices of $P_\mathbb{N}$ and $Q_\mathbb{N}$.**  Let $\varphi \in (0, 1/2)$ and $\psi \in (0, 1/2 - \varphi)$ be two parameters to be further specified later. We pick $Q_\mathbb{N} := Q_1 Q_2 \cdots$ with

$$Q_t(\{z\}) := \begin{cases} \varphi, & z = 0 \\ 1 - \varphi - \psi, & z = 1 \\ \psi, & z = 2 \end{cases} \quad (36)$$

for every $t \in \mathbb{N}$. The choice of $P_\mathbb{N}$ is given by $P_1 := Q_1$, and for $t > 1$,

$$K_P^{(t)}(z_1, \ldots, z_{t-1}, \{z_t\}) = \begin{cases} 1, & z_1 = 0, z_t = 2 \\ \varphi, & z_1 \neq 0, z_t = 0 \\ 1 - \varphi - \psi, & z_1 \neq 0, z_t = 1 \\ \psi, & z_1 \neq 0, z_t = 2 \\ 0, & \text{otherwise.} \end{cases} \quad (37)$$

$P_\mathbb{N}$ is then defined via (1), matching up with the corresponding technical assumption on conditional probability. These choices ensure in particular that under $P$, $Z_2, \ldots, Z_T$ are all almost surely 2 whenever $Z_1 = 0$ and follow the same distribution as under $Q$ conditioned on the event $Z_1 \in \{1, 2\}$.

**Computation of divergence and variational distance terms.**   For the variational distance term, we have $\hat{V}_T = 0$ if $Z_1 \neq 0$ because in that case all the conditional probabilities under $P$ and $Q$ are equal by definition. For the event $Z_1 = 0$, we can compute the variational distance term as

$$\hat{V}_T \stackrel{(6),(7)}{=} \frac{1}{T} \sum_{t=1}^{T} \left\| K_P^{(t)}(Z_1, \ldots, Z_{t-1}, \cdot) - K_Q^{(t)}(Z_1, \ldots, Z_{t-1}, \cdot) \right\|_{\mathrm{TV}}$$

$$\stackrel{(a)}{=} \frac{1}{T} \sum_{t=2}^{T} \left( K_P^{(t)}(Z_1, \ldots, Z_{t-1}, \{2\}) - K_Q^{(t)}(Z_1, \ldots, Z_{t-1}, \{2\}) \right)$$

$$\stackrel{(b)}{=} \frac{T-1}{T}(1 - \psi),$$

where in (a) we have observed that the first summand is 0 since $P_1 = Q_1$ and for the other summands, we have identified the event $Z_t = 2$ which realizes the supremum in the definition of variational distance. In step (b), we have correspondingly substituted the choices (36) and (37). For the expected variational distance term, we obtain

$$V_T = P_1(\{0\}) \cdot \frac{T-1}{T}(1-\psi) = \frac{T-1}{T}\varphi(1-\psi) \tag{38}$$

The KL divergence term can be computed as

$$D(P\|Q) \stackrel{(a)}{=} \sum_{z_1,\ldots,z_T=0}^{2} P(\{(z_1,\ldots,z_T)\}) \log \frac{P(\{(z_1,\ldots,z_T)\})}{Q(\{(z_1,\ldots,z_T)\})}$$

$$\stackrel{(1)}{=} \sum_{z_1,\ldots,z_T=0}^{2} P(\{(z_1,\ldots,z_T)\}) \log \frac{\prod_{t=1}^{T} K_P^{(t)}(z_1,\ldots,z_{t-1},\{z_t\})}{\prod_{t=1}^{T} K_Q^{(t)}(z_1,\ldots,z_{t-1},\{z_t\})}$$

$$= \sum_{z_2,\ldots,z_T=0}^{2} P(\{(0, z_2,\ldots,z_T)\}) \log \frac{K_P^{(1)}(\{0\}) \prod_{t=2}^{T} K_P^{(t)}(0, z_2,\ldots,z_{t-1},\{z_t\})}{K_Q^{(1)}(\{0\}) \prod_{t=2}^{T} K_Q^{(t)}(0, z_2,\ldots,z_{t-1},\{z_t\})}$$

$$+ \sum_{z_1=1}^{2} \sum_{z_2,\ldots,z_T=0}^{2} P(\{(z_1,\ldots,z_T)\}) \log \frac{\prod_{t=1}^{T} K_P^{(t)}(z_1,\ldots,z_{t-1},\{z_t\})}{\prod_{t=1}^{T} K_Q^{(t)}(z_1,\ldots,z_{t-1},\{z_t\})}$$

$$\stackrel{(b)}{=} P(\{(0, 2,\ldots,2)\}) \log \frac{K_P^{(1)}(\{0\}) \prod_{t=2}^{T} K_P^{(t)}(0, 2,\ldots,2,\{2\})}{K_Q^{(1)}(\{0\}) \prod_{t=2}^{T} K_Q^{(t)}(0, 2,\ldots,2,\{2\})}$$

$$+ \sum_{z_1=1}^{2} \sum_{z_2,\ldots,z_T=0}^{2} P(\{(z_1,\ldots,z_T)\}) \left( \log \frac{K_P^{(1)}(\{z_1\})}{K_Q^{(1)}(\{z_1\})} \right.$$

$$\left. + \log \frac{\prod_{t=2}^{T} K_P^{(t)}(z_1,\ldots,z_{t-1},\{z_t\})}{\prod_{t=2}^{T} K_Q^{(t)}(z_1,\ldots,z_{t-1},\{z_t\})} \right)$$

$$\stackrel{(c)}{=} K_P^{(1)}(\{0\}) \prod_{t=2}^{T} K_P^{(t)}(0, 2,\ldots,2,\{2\}) \log \frac{K_P^{(1)}(\{0\}) \prod_{t=2}^{T} K_P^{(t)}(0, 2,\ldots,2,\{2\})}{K_Q^{(1)}(\{0\}) \prod_{t=2}^{T} K_Q^{(t)}(0, 2,\ldots,2,\{2\})}$$

$$\stackrel{(36),(37)}{=} \varphi \log \frac{\varphi}{\varphi \psi^{T-1}}$$

$$= (T-1)\varphi \log \frac{1}{\psi}, \tag{39}$$

where in (a) we have used the definition of Kullback-Leibler divergence, in (b) we have used in the first summand that $P(\{(0, z_2,\ldots,z_T)\}) = 0$ unless $z_2 = \cdots = z_T = 2$, and in (c) we have used the fact that due to the definitions of $P$ and $Q$, all logarithm arguments in the second summand are 1, while in the first summand, we have applied (1).

**Computation of regret.**   It is clear from (3) and (36) (also recall $\varphi + \psi < \frac{1}{2}$) that

$$B_1 = \cdots = B_T = 1, \tag{40}$$

and it is clear from (2) and (37) that

$$B_1^* = 1, \forall t > 1 \ B_t^* = \begin{cases} 1, & Z_1 \neq 0 \\ 2, & Z_1 = 0. \end{cases} \tag{41}$$

We can calculate the average per-round regret from (4), (5), (37), (40), and (41) as

$$\Delta = \frac{1}{T} \sum_{t=1}^{T} (\ell(B_t, Z_t) - \ell(B_t^*, Z_t)) = \frac{1}{T} \sum_{t=2}^{T} (\ell(B_t, Z_t) - \ell(B_t^*, Z_t)) = \begin{cases} 0, & Z_1 \neq 0 \\ \frac{T-1}{T}, & Z_1 = 0. \end{cases} \tag{42}$$

**Choice of parameters.** We first define a sequence $(\delta_n)_{n \in \mathbb{N}}$ via $\delta_n := 1/(n+3)$ for all $n \in \mathbb{N}$. This choice ensures that the sequence takes values of at most $1/4$ and

$$\lim_{n \to \infty} \delta_n = 0 \tag{43}$$

(which are the only properties implied by this choice that we will need). We next define a sequence $(T_n)_{n \in \mathbb{N}}$ via

$$T_n := \min \left\{ T \in \mathbb{N} : \varepsilon(T, \delta_n) < \frac{1}{n} \wedge (n = 1 \vee T > T_{n-1}) \right\}.$$

This is possible since by assumption, $\varepsilon(T, \delta) \to 0$ for any fixed $\delta$ as $T \to \infty$. Again, this specific choice is only given to ensure that we have a well-defined $(T_n)_{n \in \mathbb{N}}$. The only properties we will need are that the sequence is strictly increasing and

$$\lim_{n \to \infty} \varepsilon(T_n, \delta_n) = 0. \tag{44}$$

We now specify $\psi := 1/8$, $T := T_n$ and $\delta, \varphi := \delta_n$, where the choice of $n$ is left open for now. Substituting these choices as well as the calculations (38) and (39), we can rewrite (19) and (20) as

$$\Delta \geq C \cdot \frac{7}{8} \cdot \frac{T_n - 1}{T_n} \cdot \delta_n^{1-\alpha} + \varepsilon(T_n, \delta_n) =: R_1 \tag{45}$$

$$\Delta \geq C \cdot \sqrt{\frac{T_n - 1}{T_n} \log 8} \cdot \delta_n^{\frac{1}{2}-\beta} + \varepsilon(T_n, \delta_n) =: R_2, \tag{46}$$

where we have introduced $R_1$ and $R_2$ to denote the right-hand side of these inequalities. Note that under the choices we have made in this proof, $R_1$ and $R_2$ are also equal to the right-hand sides of (19) and (20), respectively. Recalling the assumptions $\alpha < 1, \beta < 1/2$ from the theorem statement and the properties (43), (44) ensured by our construction as well as the fact that $(T_n)_{n \in \mathbb{N}}$ is chosen as a strictly increasing sequence, we observe that

$$\lim_{n \to \infty} R_1 = 0, \quad \lim_{n \to \infty} R_2 = 0, \quad \lim_{n \to \infty} \frac{T_n - 1}{T_n} = 1.$$

These limiting behaviors ensure that we are able to pick $n$ in such a way that

$$R_1, R_2 < \frac{T_n - 1}{T_n}.$$

This concludes the specification of all the parameters we have used in our construction. We can now bound (for $i = 1, 2$)

$$P(\Delta \geq R_i) \geq P\left(\Delta \geq \frac{T_n - 1}{T_n}\right) \overset{(42)}{=} \varphi = \delta_n,$$

which concludes the proof of the theorem.

## G  THE AZUMA-HOEFFDING INEQUALITY

In this appendix, we state the Azuma-Hoeffding inequality in the form in which we need it in this paper. Since we are not aware of a reference that states and proves it in this exact form, we prove it as a straightforward consequence of (Roch, 2024, Theorem 3.2.1).

**Theorem 6.** *Let $\mathscr{F} = (\mathscr{F}_t)_{t=0}^T$ be a filtration, $(X_t)_{t=0}^T$ be a supermartingale with respect to $\mathscr{F}$, and $(Y_t)_{t=1}^T, (Z_t)_{t=1}^T$ be $\mathscr{F}$-predictable processes such that for all $t \in [T]$, we have almost surely*

$$Y_t \leq X_t - X_{t-1} \leq Z_t, \quad Z_t - Y_t \leq c_t$$

*where $c_1, \ldots, c_T \in \mathbb{R}$. Then, for all $\varepsilon \in (0, \infty)$,*

$$\mathbb{P}\left(X_T - X_0 \geq \varepsilon\right) \leq \exp\left(\frac{2\varepsilon^2}{\sum_{t=1}^T c_T^2}\right).$$

*Proof.* We write $(X_t)_{t=1}^T$ in terms of its Doob decomposition (see, e.g., (Çinlar, 2011, Theorem 3.2))

$$X_t = M_t + \tilde{X}_t,$$

where $(M_t)_{t=1}^T$ is a martingale and $(\tilde{X}_t)_{t=1}^T$ is $\mathscr{F}$-predictable and nonincreasing. Then we have for all $t \in [T]$ the identity $M_t - M_{t-1} = X_t - X_{t-1} + \tilde{X}_t - \tilde{X}_{t-1}$ and hence

$$Y_t + \tilde{X}_t - \tilde{X}_{t-1} \leq M_t - M_{t-1} \leq Z_t + \tilde{X}_t - \tilde{X}_{t-1}.$$

Clearly, both $(Y_t + \tilde{X}_t - \tilde{X}_{t-1})_{t=1}^T$ and $(Z_t + \tilde{X}_t - \tilde{X}_{t-1})_{t=1}^T$ are predictable processes and we have

$$\left(Z_t + \tilde{X}_t - \tilde{X}_{t-1}\right) - \left(Y_t + \tilde{X}_t - \tilde{X}_{t-1}\right) = Z_t - Y_t \leq c_t$$

almost surely for every $t \in [T]$. We can therefore bound

$$\mathbb{P}\left(X_T - X_0 \geq \varepsilon\right) = \mathbb{P}\left(M_T - M_0 + \tilde{X}_T - \tilde{X}_0 \geq \varepsilon\right)$$

$$\overset{(a)}{\leq} \mathbb{P}\left(M_T - M_0 \geq \varepsilon\right)$$

$$\overset{(b)}{\leq} \exp\left(\frac{2\varepsilon^2}{\sum_{t=1}^T c_T^2}\right),$$

where (a) is because $(\tilde{X}_t)_{t=1}^T$ is nonincreasing and (b) is due to (Roch, 2024, Theorem 3.2.1). □

## H IMPLEMENTATION DETAILS FOR THE NUMERICAL EXPERIMENTS IN SECTION 7

Due to the summation constraint (23), we can parametrize the parameter set as

$$\Theta := \Big\{\theta \in [0,1]^{(S-1) \cdot S^m} : \theta = (p_{1|s_1,\ldots,s_m}, \ldots, p_{S-1|s_1,\ldots,s_m})_{s_1,\ldots,s_m \in \mathcal{Z}},$$

$$\forall s_1, \ldots, s_m \in \mathcal{Z} \; p_{1|s_1,\ldots,s_m} + \cdots + p_{S-1|s_1,\ldots,s_m} \leq 1\Big\}. \quad (47)$$

In this example, we do not assume any prior knowledge on the part of the learner other than the true distribution of the process is described by one of the parameters contained in $\Theta$ (i.e., it is some Markov chain with memory at most $m$). As $w$ in (22) we use a uniform distribution on $\Theta$.

Our implementation of the learner's policy (21) needs to compute

$$\mathbb{E}_{Q_{\mathbb{N}}}\left(\ell(b, Z_t)|Z_1 = z_1, \ldots, Z_{t-1} = z_{t-1}\right) = \mathbb{E}_{Q_{\mathbb{N}}}\left(\mathbb{1}_{b \neq Z_t}|Z_1 = z_1, \ldots, Z_{t-1} = z_{t-1}\right)$$

$$= 1 - K_Q^{(t)}(z_1, \ldots, z_{t-1}, \{b\})$$

for all $b \in \mathcal{B}$ and then choose a minimum. For the Markov kernel, we have

$$K_Q^{(t)}(z_1, \ldots, z_{t-1}, \{b\}) = \frac{Q_{[t]}(\{(z_1, \ldots, z_{t-1}, b)\})}{Q_{[t-1]}(\{(z_1, \ldots, z_{t-1})\})}$$

$$\overset{(22)}{=} \frac{\int_\Theta P_{\theta,[t]}(\{(z_1, \ldots, z_{t-1}, b)\})w(\theta)d\theta}{\int_\Theta P_{\theta',[t-1]}(\{(z_1, \ldots, z_{t-1})\})w(\theta')d\theta'}$$

$$= \int_\Theta \text{post}(\theta|z_1, \ldots, z_{t-1})K_{P_\theta}^{(t)}(z_1, \ldots, z_{t-1}, \{b\})d\theta, \quad (48)$$

where $P_{\theta,[t]}$ denotes the marginal of $P_\theta$ for the first $t$ components and

$$\text{post}(\theta|z_1,\ldots,z_{t-1}) := \frac{w(\theta)}{\int_\Theta P_{\theta',[t-1]}(\{(z_1,\ldots,z_{t-1})\})w(\theta')d\theta'} P_{\theta,[t-1]}(\{(z_1,\ldots,z_{t-1})\}) \quad (49)$$

clearly is a probability density function on $\Theta$. Since $w(\theta)$ does not depend on $\theta$ (as we have chosen $w$ to be a uniform density), only $P_{\theta,[t-1]}(\{(z_1,\ldots,z_{t-1})\})$ in (49) depends on $\theta$ while the fraction in front of it is a normalization factor not dependent on $\theta$. Hence, we can approximate the integration (48) using the Metropolis-Hastings algorithm (Hastings, 1970), which only requires us to evaluate the unnormalized density $P_{\theta,[t-1]}(\{(z_1,\ldots,z_{t-1})\})$ and the integrand $K_{P_\theta}^{(t)}(z_1,\ldots,z_{t-1},\{b\})$ and is generally well-behaved for high-dimensional parameter spaces $\Theta$.

## I    DISCLOSURE OF LLM USAGE

A large language model (LLM) was used for coding assistance while writing the simulation code for conducting the numerical experiments described in Section 7. The LLM was used to assist with routine programming tasks including code completions, finding bugs, and refactoring code. It was not used for development of algorithms and all code involving the generation of results reported in the paper, including Figure 1 was carefully checked by the authors.

