# OpenReview forum: "Online Prediction of Stochastic Sequences with High Probability Regret Bounds"
_ICLR.cc/2026/Conference — ICLR 2026 Poster_

### Official Review · Reviewer_eMgE · 2025-10-27

**Soundness:** 3
**Presentation:** 3
**Contribution:** 3
**Rating:** 6
**Confidence:** 4

**Summary:**

This paper studies the classical universal prediction problem as in Merhav & Feder (1998).
Here, we assume some underlying distribution (random process) $P$ that generates a sequence $Z_1, \cdots, Z_T$.
We are given an approximating distribution $Q$ of $P$, so that at each time step, the learner predicts the Bayesian-optimal decision under $Q$.
It is known (and fairly straightforward) that the expected regret between the Bayesian-optimal rule under $Q$ and the (unobservable) Bayesian strategy under $P$ is bounded above by $O(\sqrt{T \cdot D(P\|Q)})$.
This paper strengthens the result by establishing a high-probability bound of the form $O(\sqrt{T D(P\|Q)/\delta})$, where $\delta$ is the confidence parameter.
The paper further demonstrates that the $1/\sqrt{\delta}$ dependency (and for total variation, the $1/\delta$ dependency) cannot be improved if no additional structure is known between $Q$ and $P$.

**Strengths:**

1. In my opinion, the main technical contribution of the paper is the lower bound (Theorem 5), which shows that with only a global bound on the divergence between the approximate $Q$ and the underlying true $P$, the dependency on $\delta$ cannot be better than polynomial. This bound contrasts with much of the prior results in learning with expert advice and bandit settings, where the dependency is typically polylogarithmic.

2. The paper instantiates their upper bound in several settings where an approximate $Q$ is obtainable, such as countable $Z$, i.i.d., and Markov settings.

**Weaknesses:**

1. The upper bound proof is fairly simple and follows from a straightforward application of martingale concentration inequalities.

2. I'm not sure how significant this result is, as in many natural settings we already have high-probability bounds that depend logarithmically on $\delta$. The lower bound in this paper feels more like a pathological example. For instance, the examples in Section 6 only show the upper bounds; if the authors could demonstrate any **natural** settings where a polynomial dependency on $\delta$ is inevitable, the significance of the paper would be substantially boosted.

3. All the results in this paper are based on the hypothetical approximating distribution $Q$. It would be better to have some results that depend on the specific problem structure or on particular algorithms.

**Questions:**

I think the negative result in this paper is similar in spirit to the work of I. Aden-Ali, Y. Cherapanamjeri, A. Shetty, and N. Zhivotovskiy, *“The One-Inclusion Graph Algorithm is Not Always Optimal,”* in COLT 2023. It would be helpful for the authors to discuss it, as both works highlight cases where polynomial rather than logarithmic dependencies are unavoidable.

---

> ### Author Response · Authors · 2025-11-21
>
> Thank you for your thoughtful review which has helped us to significantly improve the clarity of our paper in many places, has motivated us to significantly strengthen the paper by including numerical experiments (Section 7 in the revised version), and provided us an additional literature reference that helps putting one of our results in better overall context. We think that we were able to address your concerns in the revised version of the paper. In the following, we give a detailed point-by point response to your questions and concerns. Please also see the top-level comment for an overview of all changes we have made in the revised version of the paper.
>
> > The upper bound proof is fairly simple and follows from a straightforward application of martingale concentration inequalities.
>
> We agree that the proof is not complicated, but would like to emphasize that a simple and easy to understand proof can be considered a strength rather than a weakness because it provides more and easier to understand insights into the nature of the problem and has a higher potential for reuse and generalization. While the application of the martingale concentration inequality is indeed straightforward, it can only be applied after a suitable martingale is identified. We would also like to emphasize that although the problem is classical (it is at least a quarter century old and quite well known), such a proof has not been available in the literature prior to this paper.
>
> > I'm not sure how significant this result is, as in many natural settings we already have high-probability bounds that depend logarithmically on $\delta$. The lower bound in this paper feels more like a pathological example. For instance, the examples in Section 6 only show the upper bounds; if the authors could demonstrate any natural settings where a polynomial dependency on $\delta$ is inevitable, the significance of the paper would be substantially boosted.
>
> We are not aware of an application setting where a polynomial dependence on $\delta$ is unavoidable. The contribution of this paper is that it conclusively answers the question what types of bounds can be proven *universally* for the entire class of algorithms that can be represented as in (3). We have added Remark 1 in the revised version which illustrates in what sense this class of algorithms is universal. While this does not preclude the possibility that many or even most algorithms may have tighter bounds, it does guide future research by conclusively showing that tighter bounds cannot be proven in full generality.
>
> Regarding the significance of the upper bound, we also want to emphasize that it is a significant contribution in and of itself to provide bounds that vanish at rate $\mathcal{O}(T^{-1/2})$, resp. $\mathcal{O}((T/\log T)^{-1/2})$ for every fixed $\delta$, which is not clear at all from prior research: Since instantaneous regret can be negative, the expectation bounds do not preclude the possibility that with probabilities close to $\frac{1}{2}$ the prediction algorithms have very high regret.
>
> > All the results in this paper are based on the hypothetical approximating distribution $Q$. It would be better to have some results that depend on the specific problem structure or on particular algorithms.
>
> We discuss how a suitable distribution $Q$ can be constructed in Section 6. We understand that we could have done a better job in making it clear how such a $Q$ is constructed and what role it would play in a practical algorithm. We have made the following changes in the revised version to address this concern:
> * In Section 3 (towards the end) we now state explicitly what $Q$ typically looks like.
> * We have added Section 7 which shows explicitly how $Q$ is constructed (most details are relegated to Appendix H) in the case of an example and how the policy induced by it can be (approximately) evaluated on a computer.
>
> > I think the negative result in this paper is similar in spirit to the work of I. Aden-Ali, Y. Cherapanamjeri, A. Shetty, and N. Zhivotovskiy, “The One-Inclusion Graph Algorithm is Not Always Optimal,” in COLT 2023. It would be helpful for the authors to discuss it, as both works highlight cases where polynomial rather than logarithmic dependencies are unavoidable.
>
> Thank you for this reference of which we were not previously aware. While it is not directly related to online prediction, it shows that a certain class of binary classification problems actually exhibit quite similar properties to the online prediction algorithms studied in our paper when it comes from changing the view from an in-expectation lens to a high-probability lens. We have therefore included a reference and a brief discussion right before our statement of Theorem 5.

---

### Official Review · Reviewer_imv5 · 2025-10-30

**Soundness:** 2
**Presentation:** 3
**Contribution:** 3
**Rating:** 6
**Confidence:** 3

**Summary:**

This paper studies online prediction of stochastic sequences over a known finite horizon T and derives high-probability regret bounds that complement prior in-expectation results. The authors establish a bound of order $O(T^{-1/2} \delta^{-1/2})$ holding with probability at least $1−\delta$, which mirrors the form of earlier expectation bounds (e.g., $O(T^{-1/2})$). They further prove an impossibility result showing that the dependence on $\delta$ cannot be improved without additional assumptions.

**Strengths:**

1. To the best of my knowledge, it’s the first work to provide high-probability regret bounds for the universal prediction of stochastic sequences without i.i.d. assumption or expert advice.

2. The paper is well-organized with a clear statement of problems and a structured presentation of results.

3. It is nice to see that the dependence on $\delta$ is already tight in their high probability bound.

**Weaknesses:**

1. Practical Applicability: Although the mismatched prediction results are general, their application to universal prediction relies on the construction of a suitable parametric family and may be intractable for a concrete problem.

2. This paper assumpts that the learner should know some distribution $Q$ which is assumed to be “sufficiently similar” to the unknown distribution $P$. I think it is quite a strong assumption. It would be better if the authors can add some discussions about how to learn $Q$.

3. Limited Discussion about Computation: The paper does not address how to computationally implement the proposed strategies. It may be computationally complex to apply the methods in practice.

**Questions:**

1. Can the authors provide a concrete example to construct a tractable $Q$ for a non-i.i.d. process and analyze how the regret bounds perform in practice?

2. Can the results be extended to more general settings, i.e. the loss functions are unbounded but may have heavy-tail or light-tail distribution?

3. This paper does not discuss the dependence on the constant $L$. Is it necessary to consider the improvement of $L$ and is it possible to achieve a tighter regret bound?

---

> ### Author Response · Authors · 2025-11-21
>
> Thank you for the detailed review and the many constructive questions and concerns about practical applicability, which have motivated us to substantially strengthen the paper by including an additional numerical experiments section that illustrates the applicability of the class of algorithms studied in this paper. We are confident that our revised version of the paper fully addresses your concerns. In the following, we give a detailed point-by point response to your questions and concerns. Please also see the top-level comment for an overview of all changes we have made in the revised version of the paper.
>
> > Practical Applicability: Although the mismatched prediction results are general, their application to universal prediction relies on the construction of a suitable parametric family and may be intractable for a concrete problem.
>
> This excellent observation has motivated us to substantially strengthen the paper by including a numerical experiment (Section 7 in the revised version). We have implemented the learner's policy that our paper studies for a specific case, showing that it can actually be tractable, and that it exhibits usable error bounds for this example (not hiding large constants in the asymptotic expressions). We also discuss briefly in that section how the same idea could be used to deal with other similar scenarios.
>
> > This paper assumpts that the learner should know some distribution $Q$ which is assumed to be “sufficiently similar” to the unknown distribution $P$. I think it is quite a strong assumption. It would be better if the authors can add some discussions about how to learn $Q$.
>
> There is an important shift in perspective between the mismatched and the universal prediction problem: In the universal prediction problem, we believe that $Q$ should be seen as a decription of how the learner learns about $P$ from observations of the process and uses these learnings to make predictions. It just happens to be the case that the same $Q$ that has these properties in the sense of the universal prediction problem happens to be a good approximator in the sense of the mismatched prediction problem. To illustrate this in the paper, we have given a very detailed explanation in the new Section 7 (most of the details are relegated to Appendix H) how the minimization of the expectation over $Q$ is implemented. It can be seen that the resulting expressions have a strong Bayesian flavor in that belief of what $P$ looks like is successively refined as more observations become available, but also pointed out that the theoretical results guarantee that it is not necessary for the convergence bounds to hold that the "prior" actually represents a prior belief.
>
> > Limited Discussion about Computation: The paper does not address how to computationally implement the proposed strategies. It may be computationally complex to apply the methods in practice.
>
> We have taken care to include many details and discussions on this in the new Section 7 which we are confident address this concern.
>
> > Can the authors provide a concrete example to construct a tractable $Q$ for a non-i.i.d. process and analyze how the regret bounds perform in practice?
>
> Yes, we give such an example in the new Section 7 in the revised version of the paper.
>
> > Can the results be extended to more general settings, i.e. the loss functions are unbounded but may have heavy-tail or light-tail distribution?
>
> Potentially yes, but we would expect that then the boundedness assumption needs to be replaced by additional assumptions on how the loss and the distribution $P$ interact (we would expect that just placing assumptions on $P$ would not be enough)
>
> > This paper does not discuss the dependence on the constant $L$. Is it necessary to consider the improvement of $L$ and is it possible to achieve a tighter regret bound?
>
> The constant $L$ comes from Lemma 2. Since we think it is possible to construct a pathological loss function which depends on $P$ and $Q$ but satisfies all of the assumptions in the paper and has the property that the inequalities in eq. (29) -- (31) hold with equality, we do not expect that the result of Lemma 2 (which directly enters into the main theorem) can be strengthened in terms of its dependence on $L$.

---

### Official Review · Reviewer_KtRv · 2025-11-02

**Soundness:** 3
**Presentation:** 3
**Contribution:** 3
**Rating:** 6
**Confidence:** 3

**Summary:**

This paper studies universal online prediction of stochastic sequences with a known horizon and upgrades classic in-expectation guarantees to high-probability bounds under general bounded losses. It analyzes a mismatched predictor under a reference law and then transfers the results to universal prediction via Bayesian mixtures. The main results give pathwise and distributional high-probability bounds with the same $1/\sqrt{T}$ scaling as expectation bounds, plus an impossibility result that explains the sharp dependence on the confidence parameter. The paper also summarizes consequences for countable alphabets and standard i.i.d./finite-memory Markov families.

**Strengths:**

- Clear modular proof strategy (martingale concentration around an empirical TV term, plus information-distance control) that feels broadly reusable
- First high-probability regret guarantees in this stochastic-sequence setup with general bounded loss and non-i.i.d. data
- Useful universal-prediction corollaries via mixtures; the rate summary is easy to parse.

**Weaknesses:**

- I find the dependence on the confidence parameter heavy: once the random pathwise term is de-randomized, the bounds pick up $1/\delta$ or $1/\sqrt{\delta}$ factors. I get that the impossibility result shows this is unavoidable in full generality, but for practical confidence targets the guarantees feel conservative. Readers would be curious to see whether under mild added assumptions (mixing, exp-concavity/log-loss) one can recover something closer to $\log(1/\delta)$.
- From a practicality angle, calibration seems missing. The universal consequences rely on large mixtures, which are intractable and may hide large constants. I’d feel more confident with a small synthetic study (finite-alphabet, or simple parametric families) that reports actual constants and clarifies the assumptions behind the rate table.

**Questions:**

Please address the concerns raised above.

---

> ### Author Response · Authors · 2025-11-21
>
> Thank you for your thoughtful review which has motivated us to strengthen the paper substantially by including numerical experiments for an example. We believe that we were able to address all of your concerns in the revised version of the paper. In the following, we give a detailed point-by point response to your questions and concerns. Please also see the top-level comment for an overview of all changes we have made in the revised version of the paper.
>
> > I find the dependence on the confidence parameter heavy (...)
>
> We agree that this is a limitation of the paper (which is openly discussed in the Limitations section). We have indeed considered the assumptions you proposed but do not expect that they would be sufficient to prove bounds with a more favorable dependence on $\delta$ (however, this is not conclusive). But we would like to stress that it is a significant contribution in and of itself to provide bounds that vanish at rate $\mathcal{O}(T^{-1/2})$, resp. $\mathcal{O}((T/\log T)^{-1/2})$ for every fixed $\delta$, which is not clear at all from prior research: Since instantaneous regret can be negative, the expectation bounds do not preclude the possibility that with probabilities close to $\frac{1}{2}$ the prediction algorithms have very high regret.
>
> > (...) I’d feel more confident with a small synthetic study (finite-alphabet, or simple parametric families) that reports actual constants (...)
>
> This is an excellent observation which has motivated us to substantially strengthen the paper. The main focus of this paper is to provide universal bounds that hold for large families of predictors. It is common that such generality, while it offers broad applicability due to the few assumptions it make, comes at the cost of strength of the bounds. That being said, we believe that the class of algorithms we study has the potential to provide a framework for the construction of practical prediction algorithms with theoretical high-probability guarantees. In the revised version of the paper, we show how this could be done by implementing the learner's policy we investigate (showing that it is indeed tractable in some important cases) and running numerical experiments with it. A detailed description and discussion of results can be found in Section 7 of the revised paper. The results show that at least in the case investigated in these experiments, no overly large constants seem to be hidden in the asymptotic regret convergence bounds.

---

### Official Review · Reviewer_dppa · 2025-11-06

**Soundness:** 3
**Presentation:** 3
**Contribution:** 2
**Rating:** 4
**Confidence:** 3

**Summary:**

This paper studies the universal prediction problem of stochastic sequences. Given a sequence $Z_1, \ldots, Z_{T}$, the goal is to give a prediction at every step $t$ based on the previous observations $Z_1, \ldots, Z_{t-1}$ that achieves a small error compared to the optimal strategy that has prior knowledge about the underlying distribution of the sequence. The authors show that $(T\delta)^{-1/2}$ regret can be achieved with probability $1-\delta$, and the polynomial dependence on $\delta^{-1}$ cannot be improved in the algorithm analysis.

**Strengths:**

1. Prediction of stochastic sequences is a fundamental problem, and a high-probability guarantee is often more relevant for practitioners than expected error. This work fills a gap in previous work in that only an expected regret was known.
2. The authors show that in the high-probability setting, the same convergence rate of $T^{-1/2}$ w.r.t the time horizon can be achieved. The algorithmic framework has some generality, in that similar bounds are also obtained for other settings with different assumptions on the input domain.

**Weaknesses:**

1. The main weakness is that the dependence on $\delta$ is not ideal, and it seems that the lower bound result does not hold generally for all algorithms. Specifically, Theorem 5 appears to be specific to some class of policies satisfying equation (3) in the paper. Thus, it only shows that the *error analysis* for the proposed algorithm is in a way optimal, but does not seem to be a fundamental limit that generally applies to all algorithms.

**Questions:**

1. It would be great if the authors could confirm whether Theorem 5 generally holds for all algorithms, or only algorithms that satisfy certain structures or assumptions. If it is the latter case, why would it be reasonable to focus on these algorithms?
3. For what distribution families do the authors think could achieve $log(1/\delta)$ dependence?

---

> ### Author Response · Authors · 2025-11-21
>
> Thank you for your thorough review and for raising some questions and concerns that have helped us to make important clarifications in our paper and also motivated us to strengthen the paper significantly by adding numerical experiments (Section 7) which show that the class of algorithms we study in the paper can be computationally tractable at least in some important cases. We are confident that we were able to fully address your concerns in the revised version of the paper. In the following, we give a detailed point-by-point response to your comments. Please also see the top-level comment for an overview of all changes we have made in the revised version of the paper
>
> > The main weakness is that the dependence on $\delta$ is not ideal, and it seems that the lower bound result does not hold generally for all algorithms. Specifically, Theorem 5 appears to be specific to some class of policies satisfying equation (3) in the paper. Thus, it only shows that the error analysis for the proposed algorithm is in a way optimal, but does not seem to be a fundamental limit that generally applies to all algorithms.
>
> This is an excellent observation which has prompted us to insert a crucial clarification into the paper. In fact, condition (3) is far less restrictive than it might seem. In fact, in some important cases *every* possible prediction policy has a representation of the form (3). We agree that it is very important to argue this in detail in the paper and have therefore added Remark 1 in the revised version (right after eq. (3) with technical details relegated to Appendix A) which gives full details.
>
> > It would be great if the authors could confirm whether Theorem 5 generally holds for all algorithms, or only algorithms that satisfy certain structures or assumptions. If it is the latter case, why would it be reasonable to focus on these algorithms?
>
> Theorem 5 shows that no better bound can be proven that holds for all algorithms satisfying (3). As mentioned above, we have added Remark 1 to the paper which shows that (3) is far less restrictive than it may seem.
>
> > For what distribution families do the authors think could achieve $\log(1/\delta)$ dependence?
>
> The only case we are aware of where we think such a dependence can be proved is for the case that $P$ is an i.i.d. distribution, however, since this case has marginal relevance for online prediction problems, we have decided to not include it in the paper.

---

> > ### Comment · Reviewer_dppa · 2025-11-25
> >
> > Thanks for your clarification about equation (3). You have addressed my concerns, and I am willing to increase my score to 6.

---

### Official Review · Reviewer_QjcE · 2025-11-06

**Soundness:** 3
**Presentation:** 3
**Contribution:** 3
**Rating:** 8
**Confidence:** 3

**Summary:**

This paper revisits the classical problem of universal prediction of stochastic sequences and aims to complement existing in-expectation regret bounds in (Merhav and Feder, 1998) by deriving high-probability bounds. The setting involves a learner predicting stochastic outcomes over a known finite horizon with losses measured by a bounded function. They show $O(T^{-1/2}\delta^{-1/2})$ convergence rate with probability at least $1-\delta$, compared to the classical $O(T^{-1/2})$ rate in expectation. The authors also establish an impossibility result, proving that the bounds cannot be improved significantly without stronger assumptions.

**Strengths:**

- The paper studies a fundamental problem.
- The paper is well-written and has a clear related-work section.
- The paper presents an impossibility theorem clarifying the optimality of the proposed dependence on $\delta$

**Weaknesses:**

- The paper heavily focusses on theoretical analysis while leaving numerical experiments as future research directions.
- The related work section could be tightened. Citations to multi-armed bandit and MDP literature feel tangential since those problems involve decision-making and exploration-exploitation tradeoffs whereas the present paper studies passive sequence prediction. Such citations distract from the main focus on universal prediction although there might be possible methodological overlap.

**Questions:**

- The paper extends classical expected regret guarantees to high-probability bounds. Could the authors better illustrate practical scenarios or domains (e.g., time-series prediction, online compression, or universal coding) where such high-probability guarantees provide tangible benefits over expected bounds?

---

> ### Author Response · Authors · 2025-11-21
>
> Thank you for your encouraging review and the constructive feedback which has helped us a lot to further improve the paper. In particular, it has motivated us to strengthen the paper substantially by including numerical experiments for an example (Section 7 in the revised version). In the following, we give a detailed point-by point response to your questions and concerns. Please also see the top-level comment for an overview of all changes we have made in the revised version of the paper.
>
> > The paper heavily focusses on theoretical analysis while leaving numerical experiments as future research directions.
>
> We agree that this is a significant limitation, even for a paper that has a strong focus on theoretical results. We have therefore decided to include numerical experiments for predicting a Markov chain with memory in the revised version of the paper (Section 7 in the revised version, look for the blue text).
>
> > The related work section could be tightened. (...)
>
> As recommended, we have removed the pars of the literature survey that deals with multi-arm bandits and the parts of the introduction that deal with Markov decision processes and bandits.
>
> > (...) Could the authors better illustrate practical scenarios or domains (e.g., time-series prediction, online compression, or universal coding) where such high-probability guarantees provide tangible benefits over expected bounds?
>
> We will focus on the domain of time-series prediction which is the one we were mostly motivated by when writing the paper. Since instantaneous regret can be negative, an expected regret bound does not by itself guarantee that the predictions made by a learner are as accurate as they can be most of the time -- in principle, it is possible that half of them are catastrophically bad and the other half are extremely good. For safety-critical applications it is often not acceptable that the prediction quality of a significant fraction of the time series that the algorithm is deployed to is bad. Examples of such safety-critical scenarios for which specific ML algorithms have been developed and evaluated specifically for reliability include accident prediction in air traffic control (Amin et al., 2024), safety verification in autonomous driving (Zhang et al., 2022), and sepsis prediction in healthcare (Boussina et al., 2024). Our paper focuses more fundamentally on a universal algorithm that is not specific to any of these problems, but we think that there is long-term potential for such universal algorithms to be valuable tools for developing algorithms for more specific applied problems.
>
> This is an excellent question, and we believe that answering it also in the paper will strengthen its overall motivation. We have therefore included these three references in our introduction before the list of contributions (look for the blue text).

---

### Author Response · Authors · 2025-11-21
**Summary of main changes in the revised version**

We are very grateful to the reviewers who have taken the time and effort to write very thoughtful reviews with many detailed comments that have motivated and helped us to substantially strengthen our paper as well as improve its readability. We think that we were able to address all questions and concerns raised by the reviewers. In the following, we summarize the main changes that we have made in the revised version:

* We have added numerical experiments (Section 7 in the revised version) to show that the learner's policy we analyze in the paper can indeed be implemented and to illustrate that it does incorporate a formalized way to learn from past observations.
* We have added Remark 1 in the revised version which illustrates that the class of learner's policies we study in this paper actually contains *every possible policy* in some important cases (and we believe that the argument we make for this also illustrates that the class is far less restrictive than it might appear in more general cases).
* We have added many additional clarifications that address questions and concerns raised by reviewers and substantially improve the readability and strength of the paper. Please see our individual responses to the reviewers for details.

An LLM was used to assist with routine programming tasks when conducting the numerical experiments. As per ICLR policy, the precise role of the LLM is disclosed in Appendix I of the revised version of the paper.

---

### Author Response · Authors · 2025-12-01
**Summary of review/rebuttal process and final remarks**

The constructive criticisms from the reviewers have helped us to **significantly further improve** what was already a highly rated paper. In particular:

* The most frequently raised (Reviewers QjcE, KtRv, imv5, eMgE) and most strongly emphasized concern was that we do not sufficiently address practical applicability issues in our original submission. We have comprehensively addressed this issue by adding a new section in the revised version which explains in detail how the algorithm works in the case of an example, we report numerical results, we discuss the computational complexity both of the examples and of similar but more complex problems, and how computational complexity issues for larger examples can be mitigated. Unfortunately, none of these four reviewers had a chance to respond before the discussion period was cut short. However, we are certain that we were able to comprehensively address this main concern about our paper which was already highly rated before.
* There were some concerns about the generality and applicability of our framework, particularly emphasized by reviewer dppa. In response to this observation, we have added an additional discussion including a mathematical argument (Remark 1 and Appendix A in the revised version) to show that the framework is indeed more general and more widely applicable than it might seem. The reviewer has acknowledged during discussion that their concern was fully addressed with this change and raised their score accordingly.
* There were several minor concerns from reviewers about technical strength and applicability of our results which we believe were a result of misunderstandings. We acknowledge that these misunderstandings were caused by several minor flaws in our explanations and discussions. Consequently, we have taken the opportunity to improve our presentation and discussion in several places (see our point-by-point responses below). Unfortunately, the reviewers didn't have a chance to respond before discussions were cut short, but we are confident that we have addressed every concern raised about what was already a highly rated paper.

We would also like to emphasize that even before our substantial revisions, **the reviewers' view of our paper was very positive**. In particular, they have emphasized the following strengths of our paper:

* The problem we consider is of fundamental importance and interest in the field (specially emphasized by reviewers QjcE, dppa)
* Our results are novel (especially emphasized by reviewers dppa, KtRv, imv5)
* The results are useful and significant, with clear impacts both for theoretical and practical works (specially emphasized by reviewers dppa, KtRv, eMgE)
* The paper is well-written, easy to understand, and the proofs are clear and modular (specially emphasized by reviewers QjcE, KtRv, imv5)
* Our proof techniques show promise for reuse in other contexts (specially emphasized by reviewer KtRv)
* Our impossibility result shows that our analysis is optimal in the sense that the result cannot be improved without further assumptions (specially emphasized by reviewers QjcE, eMgE, imv5)

Overall, **every reviewer has indicated in their scoring** (four out of five during the initial review and the remaining one during discussion) **that our paper is clearly above the acceptance threshold** of the conference.

---

### Meta-Review · Area_Chair_jRxr · 2026-01-05

**Summary:**

This paper establishes a high-probability bound for the mismatch prediction problem. The reviewers found the problem to be interesting and fundamental, and agreed that the provided lower bound effectively demonstrates the necessity of a polynomial dependence on $1/\delta$. In the response, the authors presented the requested numerical experiments to complement the theory. Therefore, my recommendation is acceptance.

**Reviewer Concerns:**

Concerns addressed by the rebuttal:
1) the numerical experiments;
2) clarification of the related works;
3) the necessary condition to construct the hard instances (i.e., explanation of equation (3)).

Concerns which are still outstanding:
1) The assumption of known $Q$ is overly restrictive;
2) The computational cost is not clear in general case.

**Reviewer Scores:**

I expect Reviewer dppa would have increased his score to 6 (as he claimed in the discussion), and other reviewers would have kept their score assuming full engagement in the discussion section.

---

### Decision · Program_Chairs · 2026-01-26

Accept (Poster)